# LMRL Gym: Benchmarks for Multi-Turn Reinforcement Learning with Language Models

## Abstract

Large language models (LLMs) provide excellent text-generation capabilities, but standard prompting and generation methods generally do not lead to intentional or goal-directed agents and might necessitate considerable prompt tuning. This becomes particularly apparent in multi-turn conversations: even the best current LLMs rarely ask clarifying questions, engage in explicit information gathering, or take actions now that lead to better decisions after multiple turns. Reinforcement learning has the potential to leverage the powerful modeling capabilities of LLMs, as well as their internal representation of textual interactions, to create capable goal-directed language agents. This can enable intentional and temporally extended interactions, such as with humans, through coordinated persuasion and carefully crafted questions, or in goal-directed play through text games to bring about desired final outcomes. However, enabling this requires the community to develop stable and reliable reinforcement learning algorithms that can effectively train LLMs. Developing such algorithms requires tasks that can gauge progress on algorithm design, provide accessible and reproducible evaluations for multi-turn interactions, and cover a range of task properties and challenges in improving reinforcement learning algorithms. Our paper introduces the LMRL-Gym benchmark for evaluating multi-turn RL for LLMs, together with an open-source research framework containing a basic toolkit for getting started on multi-turn RL with offline value-based and policy-based RL methods. Our benchmark consists of 8 different language tasks, which require multiple rounds of language interaction and cover a range of tasks in open-ended dialogue and text games.

## 1 Introduction

Large language models (LLMs) have demonstrated remarkable abilities when naturally conversing with humans (OpenAI, 2023; 2022; Touvron et al., 2023; Google, 2023), answering questions and responding to requests (Shuster et al., 2022b;a; Qin et al., 2023), and even performing coding tasks (Chen et al., 2021; Wang et al., 2023). Much of these capabilities are enabled by learning to emulate humans from large datasets of text from the web (Völske et al., 2017; Shuster et al., 2022a; Yao et al., 2023), learning from examples "in context" (Brown et al., 2020), as well as other sources of supervision such as instruction datasets (Mishra et al., 2022; Wei et al., 2022; Wang et al., 2022b) and preference fine-tuning with RLHF (Ziegler et al., 2020; Ouyang et al., 2022). However, directly applying LLMs in settings that require planning or multi-turn interactions reveals certain difficulties. LLMs are not explicitly goal-directed, as they are not optimized to directly solve particular tasks, but rather to produce text that resembles the distribution of human-

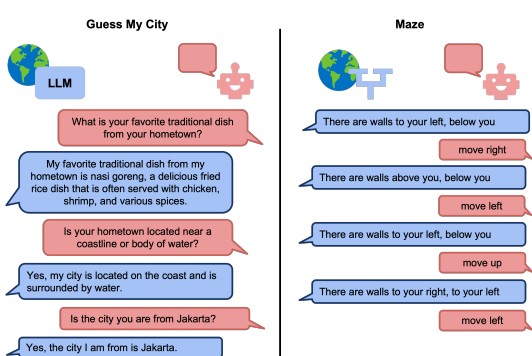

Figure 1: Overview of LMRL-Gym: We show sample trajectories from two tasks in our benchmark. In Guess My City task, the agent learns to ask questions to guess the city the oracle is from while in the Maze task, the agent learns to make the correct moves based on cues from the oracle.

provided examples or accords with human preferences (Ziegler et al., 2020; Stiennon et al., 2020; Wu et al., 2021; Bai et al., 2022a). This can become particularly apparent in temporally extended tasks, such as multi-turn dialogue (Irvine et al., 2023; , FAIR), complex tool use (Wang et al., 2022a), multi-step games (Hendrycks et al., 2021b), as well as other interactive applications. In principle, LLMs should contain the knowledge necessary to succeed in such settings: if the multi-turn interactions center around problem domains that are well represented in the model's training data (such as dialogue), well-trained LLMs should already serve as powerful predictive models in such settings.

However, leveraging this predictive knowledge to derive effective actions and strategies requires not just emulating humans, but also planning and optimization.

Multi-turn reinforcement learning (RL) (Sutton & Barto, 2018) in principle offers a path to enable LLMs to overcome challenges in goal-directed reasoning and planning in interactive, multi-turn settings, including complex dialogue, games, and tool use. We might hypothesize that RL could serve as a powerful tool for LLM training, not just for training models to accord with human preferences, but more generally to accomplish tasks in an intentional and goal-directed manner. Text generation can be viewed as a sequential decision-making process, treating a sequence of tokens as a trajectory. Many tasks, such as successfully answering questions or eliciting a desired reaction from a user, can then be framed as optimizing some reward function over these trajectories. However, despite extensive interest in RL for LLMs in recent years, much (though not all) of the recent research in this area has focused on "single-step" RL problems, where a single response is optimized for some quality metric, typically derived from human preference signals (Stiennon et al., 2020; Ziegler et al., 2020; Ouyang et al., 2022; Bai et al., 2022a; Anthropic, 2023; Ramamurthy et al., 2023; Christiano et al., 2023; Casper et al., 2023).

While some works have sought to apply RL for multi-turn tasks (Singh et al., 1999; Li et al., 2016; Shah et al., 2016; Kwan et al., 2022), particularly in the domain of goal-directed dialogue (Lewis et al., 2017; Verma et al., 2022), there has been comparatively little research on actually improving the underlying RL algorithms and very little head-to-head comparison on the same set of tasks. This is perhaps unsurprising: it is likely much easier to evaluate improvements to algorithms for single-turn text generation as compared to multi-turn generation since multi-turn dialogue requires an interactive evaluation procedure, where there is no established protocol and the "gold standard" constitutes costly and time-consuming studies with human participants.

In this work, we aim to address this challenge and make it possible for *RL algorithms researchers* to iterate on developing better RL methods for multi-turn language-based interaction tasks, such as dialogue and games. We posit that benchmarking RL algorithms for LLMs presents a very different set of challenges and merits a different set of solutions compared to other benchmarks in NLP. While most NLP benchmarks are based on standard supervised machine learning paradigms, with a training set and a test set (Marcus et al., 1993; Tjong Kim Sang & De Meulder, 2003; Socher et al., 2013; Rajpurkar et al., 2016; Wang et al., 2019; Williams et al., 2018), RL benchmarks require simulators that the trained agents can interact with to measure their performance. Until recently, constructing high-fidelity simulators for dialogue with humans has been exceptionally difficult. However, the very same LLMs that we advocate to train with RL can offer a solution here, as they can be trained to emulate human-like dialogue and produce both synthetic datasets and synthetic simulated evaluation protocols, where an RL-trained LLM agent "talks" to another LLM provided as part of the benchmark as a "simulator." While this approach to benchmarking interactive dialogue has significant downsides (e.g., the responses of the benchmark LLM might often deviate drastically from human behavior), we believe that the corresponding upside of enabling accessible benchmarking of multi-turn RL algorithms provides a compelling solution. We emphasize however that our goal is *not* to utilize this approach to benchmark whether LLMs are *good at talking to humans*, but rather as a way to test RL algorithms with datasets that are sufficiently difficult and complex so as to gauge how effective they might be *if they were then trained on data from real humans*.

Our proposed benchmark, LMRL-Gym, consists of 8 tasks that each come with an offline dataset that can be used for offline RL training, and a "simulator" that can be used to evaluate the resulting agents in terms of their performance on multi-turn interactive tasks. This simulator supports both evaluation of agents trained with offline RL, and online training. Three of these tasks are Interactive Dialogue tasks designed to simulate real-world interactions with humans requiring information gathering from humans (20 Questions, Guess My City) and negotiation (Car Dealer). Five of these tasks are RL Capability Tests designed to isolate specific desirable properties of training RL with language. In addition to presenting the benchmark tasks and the synthetic data generation system, we also provide a research framework that provides a toolkit for researchers and practitioners to get started with multi-turn RL for LLMs. This framework includes implementations of PPO (Schulman et al., 2017), ILQL (Snell et al., 2022a), and several baseline methods, implemented in an extensible way designed for future development of tasks, experimentation, and algorithm design.

## 2 RELATED WORKS

**Datasets, benchmarks, and libraries.** Benchmarks and datasets have been an important factor for driving progress in NLP in domains that include machine translation (Tiedemann, 2012; Bojar et al., 2016), natural language understanding (Rajpurkar et al., 2016; Wang et al., 2019; Hendrycks et al., 2020; 2021a; Ramamurthy et al., 2023), and solving math problems (Cobbe et al., 2021). However, these tasks generally do not involve multi-turn interaction and do not come with rewards, making them hard to adapt to RL research. For example, the standard for evaluating dialogue agents has been to run a human subjects study, but this is time-consuming and costly. Some works have proposed text games for evaluating language-based agents (Hausknecht et al., 2019; Hendrycks et al., 2021b;

Wang et al., 2022a; Yao et al., 2023). Our aim is to cover a variety of problem settings that reflect challenges in open-vocabulary interaction in addition to text games. Motivated by successes in using LLMs to generate synthetic data (Hausknecht et al., 2019; Park et al., 2023; Bai et al., 2022b), our proposed tasks are based on synthetic data. While such data may differ from natural text, the scope of our benchmark is specific to evaluating RL *algorithms*, not the ability to interact with humans.

**RL for language models.** RL for language models has seen success in aligning LLMs with human preferences (RLHF) (Ziegler et al., 2020; Stiennon et al., 2020; Bai et al., 2022a;b; Ouyang et al., 2022; Christiano et al., 2023), optimizing non-differentiable objectives for machine translation (Wu et al., 2016; Nguyen et al., 2017; Kiegeland & Kreutzer, 2021), generation (Tambwekar et al., 2019; Pang & He, 2021; Pyatkin et al., 2022), dialogue (Cuayáhuitl et al., 2015; Georgila & Traum, 2011; Li et al., 2016), question answering (Pyatkin et al., 2022), and summarization (Paulus et al., 2017; Böhm et al., 2019; Wu & Hu, 2018). These include RL methods that learn by directly interacting with the environment (online RL) (Carta et al., 2023) and RL methods that only use a static dataset (offline RL) (Jaques et al., 2020; Snell et al., 2022a; Jang et al., 2022; Verma et al., 2022; , FAIR). However, many of these works operate in the singe-step bandit setting, and do not consider multi-turn goal-directed tasks. Our benchmark, on the other hand, focuses on tasks involving multiple turns of interaction with clearly defined goal-based reward functions.

**Capabilities of LLMs.** There has been a surge in the capabilities of LLMs for generation (Ghazvininejad et al., 2017; Radford et al., 2019), dialogue (Lewis et al., 2017; Jaques et al., 2017; Shuster et al., 2022b; Snell et al., 2022b), question answering (Pyatkin et al., 2022), summarization (Paulus et al., 2017; Böhm et al., 2019; Wu & Hu, 2018), text-based games (Narasimhan et al., 2015; Hausknecht et al., 2019), translation (Gu et al., 2017), and more. However, these are often supervised learning tasks that do not test the LLMs' abilities to achieve a specific long-term objective. Research on dialogue generation (Jaques et al., 2017; He et al., 2018; Shuster et al., 2022b;a) has often focused on generating feasible-looking agent dialogue without explicit consideration for some multi-turn objective. Our benchmarks allow for the development of algorithms that enable LLMs to *interact* with an environment to achieve long-term objectives, by providing tasks with online simulators and offline datasets.

# 3 MULTI-TURN GENERATION WITH RL AND LANGUAGE MODELS

This section introduces the conceptual foundations of using reinforcement learning for multi-turn generation with language models. We introduce a definition of the Markov Decision Process for language and a framework for the methods we focus on in this paper.

## 3.1 DEFINITIONS

We formalize language generation tasks as a partially observable Markov decision process. We can think of the state as the history of tokens and action as the next token generated by the model. An observation is a single token $s_i$ in the history. The probability of generating the next token given the previous observation token $P(s_{i+1}|s_i)$ is non-Markovian. However, a Markovian state can be formed by concatenating all of the previous tokens.

A policy $\pi$ defines the agent's behavior by taking in the current state $s$ given by $[s_0, \ldots, s_i]$, and outputting a new action token $a$ given by $s_{i+1}$. The environment assigns a reward $r(s, a)$ based on the entire sequence of tokens so far. The tokens in the state are either generated by the policy $\pi$ or the environment. For example, in the Car Dealer task, the policy generates the tokens for the Seller and the environment generates the tokens for the Buyer and the history of their conversation would form the state.

A complete sequence of tokens will also be referred to as a trajectory. The goal of RL is to produce a policy $\pi^*$ that maximizes the expected discounted sum of rewards over trajectories ($\tau$) under the policy $\pi^* = \arg\max_\pi \mathbb{E}_{\tau \sim \pi} \left[ \sum_{t=0}^{T-1} \gamma^t r_t(s_t, a_t) \right]$, where $\tau$ represents a trajectory.

## 3.2 RL ALGORITHMS

Several possible RL algorithms could be used (Jaques et al., 2020; Verma et al., 2022; Snell et al., 2022a; Schulman et al., 2017; Stiennon et al., 2022; Bai et al., 2022a; Casper et al., 2023). Policy gradient methods, such as PPO (Schulman et al., 2017), directly compute the gradient of the language model concerning the expected reward objective. Value-based methods estimate a state-action ($Q$) or state-value ($V$) function, from which they then derive a policy by either acting greedily with respect to the Q-function or by combining the learned $Q$-function with the base LM by perturbing the base model's logits with the learned action-value functions (Snell et al., 2022a).

RL methods for training LLMs can be *online* or *offline*. Online methods repeatedly interact with the environment, collecting additional data during training. Offline RL instead learns to extract the best behaviors from an existing, potentially suboptimal dataset. Due to the large amount of existing text interactions on the internet, offline RL is an ideal setting for training language models.

Therefore, our work primary focuses on benchmarking offline RL algorithms. However our tasks also fully support online RL and we include an online PPO baseline in our evaluation.

# 4 THE LMRL-GYM: SYNTHETIC BENCHMARKS FOR RL WITH LANGUAGE

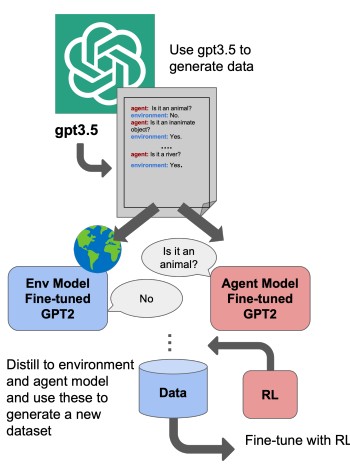

Figure 2: To generate data for conversational tasks, we use LLMs as "simulators" for the task. Our simulators can be used to generate offline data, to provide a "simulation environment" for evaluation, to perform online training, and to compute rewards.

Our benchmark consists of 8 tasks, split into two categories; RL Capability Tests and Interactive Dialogue tasks. The RL Capability tests focus on specific desirable capabilities for RL algorithms to have such as strategic decision-making, credit assignment, or trajectory stitching. For the interactive dialogue tasks, we model them after real-world interactions with humans, such as persuading someone to buy a car or playing a guessing game like 20 questions. In such multi-turn interactions, there is a need for the agent to make inferences about persuasive strategies and social interaction, actively gather information efficiently through asking questions, and strategically reason in partially observable settings. Below, we discuss the specific capabilities of RL algorithms for LLMs that our benchmark aims to evaluate, we summarize the data generation and simulation process, and we describe the tasks themselves. We provide a concise summary of the dataset and task statistics in Section 4.3.

## 4.1 EVALUATING CAPABILITIES ENABLED BY RL

A central objective of our benchmark is to evaluate the core capabilities that RL can enable in large language models. Some of these capabilities are *computational*, and relate to core decision-making irrespective of the considerations of natural language, such as playing chess, while others are semantic. We discuss the particular capabilities of algorithms we aim to evaluate in this section, followed by a discussion of the criteria that such tasks must meet to do so.

**Strategic decision making.** RL shines in goal-directed tasks that require multi-step planning and strategic decision-making. Strategic decision-making can range from simple choices like asking follow-up questions to gather information (e.g., in the 20 Questions task), to complex strategy in chess.

**Complex language.** Our benchmark includes realistic language and interaction scenarios, requiring LLMs to combine their knowledge from pretraining to help solve tasks during RL finetuning. Rather than focusing entirely on simple causal logic and strategy of the sort found in text games, several of our tasks specifically emphasize the use of realistic language.

**Credit assignment.** In RL, rewards are often delayed relative to the action that was pivotal to the outcome. For example, a seller agent might state a particularly compelling feature of the product and then, several turns later, complete a successful sale. RL must determine the statements that led to the good outcome, and reinforce them.

**Partial observability.** In language tasks, the state consists of the entire history of tokens, and an agent may need to examine this entire context to infer the correct state. For example, the mental states of a speaker in a dialogue (e.g., whether the buyer is impatient in a selling task), previously observed facts in a guessing game, and other hidden variables might induce partial observability.

**Trajectory stitching.** In a dataset with many suboptimal trajectories, it is necessary to join optimal actions from different suboptimal trajectories together to form the most optimal trajectory. An algorithm capable of trajectory stitching should be able to learn from optimal actions taken in unsuccessful trajectories and avoid suboptimal actions that occurred in successful trajectories.

## 4.2 LMRL-GYM TASKS

We split our tasks into 5 "RL Capability Tests" and 3 "Interactive Dialogue" tasks. The RL Capability tests are designed to create comparisons between tasks based on specific RL capabilities. In addition, they are chosen to be text-based versions of tasks where RL without language is known to excel. The interactive dialogue tasks are chosen to be illustrations of real-world interactions where we expect multi-turn RL to shine. Example trials for each task are shown in Figure 4.

### 4.2.1 THE RL CAPABILITY TESTS

The RL Capability Tests are text-based games designed such that they 1) isolate specific RL capabilities and 2) are language analogs of tasks where RL is known to succeed. To emphasize the comparison to a non-text-based version, we evaluate the Maze task in a symbolic or grid-based environment seen in Appendix G. Below we highlight how each of the tasks isolates the RL Properties. Further details on task design for each of the RL Property test tasks can be found in Appendix A.

|  | Strategic Decision Making | Complex Language | Credit Assignment | Partial Observability | Trajectory Stitching |
|---|---|---|---|---|---|
| Maze FO | ✗ | ✗ | ✓ | ✗ | ✓ |
| Maze PO | ✗ | ✗ | ✓ | ✓ | ✓ |
| Text-Nav FO | ✗ | ✓ | ✓ | ✗ | ✓ |
| Text-Nav PO | ✗ | ✓ | ✓ | ✓ | ✓ |
| Wordle | ✓ | ✗ | ✗ | ✓ | ✓ |
| Chess | ✓ | ✗ | ✓ | ✗ | ✓ |
| Endgames | ✓ | ✗ | ✓ | ✗ | ✓ |
| 20Qs | ✓ | ✓ | ✓ | ✓ | ? |
| Guess My City | ✓ | ✓ | ✓ | ✓ | ? |
| Car Dealer | ✓ | ✓ | ✓ | ? | ✓ |

Figure 3: Each task in the text-based games and RL Capabilities test suite is chosen to test some subset of the RL Capabilities outlined in Section 4. We use the number of checkmarks to highlight which tasks emphasize a property more than others. As can be seen in the figure, the RL Property Test tasks Chess, Endgames, Wordle, Maze, and Text-Nav all test some proper subset of all the properties. The Interactive Dialogue tasks test many of the properties, but since they were designed to be realistic rather than an RL Property Test, we are unsure of the extent to which they test some properties.

**Partial Observability.** We focus on the effect that partial observability has on performance by including both fully observed (FO) and partially observed (PO) versions of the Maze and Text-Nav tasks. We create a partially observed Maze or Text-Nav by removing information about the location from the state.

**Trajectory Stitching.** All of the RL Capability Tests test trajectory stitching, because they include suboptimal data. The inclusion of suboptimal requires an offline algorithm to utilize information from suboptimal data to generate optimal trajectories. Further details about our dataset generation strategies can be found in Appendix C.

**Credit Assignment.** Chess, Endgames, Maze and Text-Nav test credit assignment, because success in the task is dependent on things outside the control of the agent. In Chess and Endgames, victory is highly dependent on the actions of the opponent. If the opponent makes bad moves, victory is far more likely than if the opponent makes excellent moves. Similarly, in the Maze and Text-Nav tasks, the dataset is generated such that trajectories that start close to the goal are far more likely to succeed. Therefore the agent must learn to distinguish between lucky wins and those resulting from complex strategic decisions.

**Complex Language** We include both Maze and Text-Nav tasks to highlight the differences between a maze task with and without complex or stochastic text. However, the RL Capability tests are not designed primarily with complex language in mind and we leave that problem to the Interactive Dialogue tasks.

**Strategic Decision Making** Wordle, Chess, and Chess Endgames test strategic decision-making to varying degrees. Wordle tests information gathering and strategy in a partially observed environment, because the full information about the state is not provided to the agent. We find that Chess requires the most strategic decision-making because it requires the agent to plan over a game more than 40 moves in length.

**Why include Endgames (Theoretical Chess Endgames)?** Chess endgames provide a simpler and more goal-directed variation of the chess task. By focusing on the endgame, we encourage algorithms to learn strategy rather than memorizing the opening moves of a chess game. A classic theoretical endgame position consists of a position where the only pieces on the board are the two kings and the queen. Although the board position appears simple, a sequence of carefully calculated moves is required to win. A simpler board state allows language models to make progress without fewer computational resources.

### 4.2.2 INTERACTIVE DIALOGUE TASKS

For the interactive dialogue tasks, we chose two tasks that involve rational decision-making (20Qs, Guess) and information gathering and one that involves negotiation (Car Dealer). These tasks aim to simulate real world interactions between humans.

**20Qs (Twenty Questions).** This task tests information gathering to see if a policy can successfully reason about an unknown subject based on context to determine what it is. Additionally, it also evaluates the ability of the model to understand semantics, as it also needs knowledge about the objects in question. In twenty questions, one player (the oracle) thinks of an object, and the agent (the guesser) tries to guess what it is by asking a series of yes-or-no questions. In this interaction, the oracle serves as the environment, and the agent learning a policy to solve the game is the guesser.

**Guess (Guess My City).** This task simulates a more complicated guessing game, where one player (the oracle) is from a specific city, and the other player (the guesser) tries to guess what city the oracle is from. Here, the guesser can ask not only yes and no questions, but can also ask open-ended questions. This task tests strategic decision-making and the ability of algorithms to handle complex language, as it allows the agent to go beyond learning to ask yes/no questions and learning to ask questions open-ended questions that provide the agent with more information.

**Car Dealer.** This task simulates a conversation between a car buyer and a car dealer, each with different strategies for getting the best deal. The buyer wants to buy a certain type of car within a certain budget, and the car dealer wants to complete the sale ideally with a high sale price. We have designed the task such that there exist three different kinds of sellers and three different buyers, each primed with a different strategy. Hence, agents should learn to make agreements with buyers who are most compatible with their strategy. This allows us to test the ability of RL algorithms to learn strategic decision-making and credit assignment, by learning which strategies led to a successful sale of the car.

---

**– Chess –**
**environment:** r n b q k b n r / p p p p p p p p / 8 / 8 / 8 / 8 / P P P P P P P P / R N B Q K B N R  w  K Q k q  -  0  1
**agent:** N f 3

---

**– Guess My City –**
City: Jakarta, Indonesia
**agent:** What is your favorite traditional dish from your hometown?
**environment:** My favorite traditional dish from my hometown is nasi goreng, a delicious fried rice dish that is often served with chicken, shrimp, and various spices.
**....**
**agent:** Is your hometown located near a coastline or body of water?
**environment:** Yes, my city is located on the coast and is surrounded by water.
**agent:** Is the city you are from Jakarta?
**environment:** Yes, the city I am from is Jakarta.

**– Partially Observed Maze –**
**environment:** There are walls to your left, below you
move right
There are walls above you, below you
move left
There are walls to your left, below you
move up
There are walls to your right, to your left
**agent:** move up

**– Twenty Questions –**
Word: River
**agent:** Is it an animal?
**environment:** No.
**agent:** Is it an inanimate object?
**environment:** Yes.
**agent:** Is it found outside?
**environment:** Yes.
**....**
**agent:** Is it a river?
**environment:** Yes.

**– Text-Nav –**
**environment:**
-= Living Room =-
You are carrying nothing on you right now. You make out a table. You wonder idly who left that here. The table is standard. On the table you see a stale food. There are unblocked exits to the east, south and west.
Available actions: ['pick up stale food', 'go east', 'go south', 'go west']
**agent:** pick up stale food

**– Fully observed maze –**
**environment:** The goal is at position 8, 6. Your current position is at position 5, 3. There are walls to your left, below you.
**agent:** move up
**environment:** The goal is at position 8, 6. Your current position is at position 6, 3. There are walls to your left.
**agent:** move up

**– Car Dealer –**
**agent:** Good morning! Welcome to our car dealership. How can I assist you today?
**environment:** Hi. I'm looking for a Honda SUV with a backup camera. Can you show me what you have?
**....**
**agent:** Of course! We have a great selection of Honda SUVs with backup cameras. Let me show you our top models. This one here is the Honda CR-V. It has a powerful engine, spacious interior, and advanced safety features. Would you like me to go over the details?
**environment:** No, thank you. I already know about the features. Can you tell me the price?
**agent:** Alright, thanks for your time. I'll think about it.

---

Figure 4: Example trials for tasks in LMRL-Gym. Each task requires the agent to perform a multi-turn interaction with an environment – either a text game or another LLM simulating a human speaker. Full details of tasks are provided in Appendix C.

### 4.3 AN OVERVIEW OF DATA COLLECTION FOR LMRL-GYM TASKS

In order to make the tasks in LMRL-Gym practical for benchmarking RL methods, we must balance accessibility and realism. Real-world datasets would be the most realistic, but since RL algorithms need to be evaluated by actually running the learned policy, real-world tasks are comparatively inaccessible for rapid iteration (e.g., if they require talking to real humans). We therefore use simulators

for all of our tasks, which are derived either from text-based games, or conversational agents powered by language models. Although this fully synthetic setup sacrifices some realism, we believe the significant gain in accessibility is worthwhile and will enable more rapid RL algorithm progress.

**Text-based games.**  For each of these tasks, we use a simulator such as a chess engine or maze solver to generate near-optimal data and then we dilute the policy with suboptimal data by taking suboptimal actions or using inferior policies. We also convert our task from a symbolic version to a text-based version in a programmatic way as discussed in **??**.

**Conversational tasks.**  For conversational tasks, we leverage existing LLMs to generate our data, either with two instances of LLMs "talking" to one another or all at once through few-shot prompting. In order to train these LLMs, we use OpenAI's GPT-3.5 to generate an initial dataset by asking reasonable questions and answers out-of-the-box, collecting a dataset of differing sizes depending on the task. In the case of 20Qs and Guess My City, we collected 1K conversations by querying GPT-3.5 (text-davinci-003) to generate both sides of the conversation. To generate the dataset for training our algorithms, we fine-tuned a GPT2-medium guesser model and a GPT2-medium oracle model on their respective sides of the conversation. Using these models, we generated the final 100K conversations dataset by having the two models talk to each other. We conducted a similar process for the Car Dealer task.

In Table 4.3 we present the dataset statistics on these tasks. The number of trajectories and the average length of the trajectories varies based on the complexity of the tasks.

| Task | Maze | Text-Nav | Wordle | Chess | Endgames | 20Qs | Guess | Car |
|---|---|---|---|---|---|---|---|---|
| Size | 1.24k | 2.5k | 1m | 625k | 97.756k | 100k | 100k | 19k |
| avg length | 19.7 | 12.2 | 4.82 | 46.7 | 11.9 | -14.9 | -18.8 | 16.5 |
| std length | 24.5 | 8.77 | 1.27 | 18.16 | 12.0 | 4.38 | 4.57 | 3.61 |
| success rate | 0.11 | 0.26 | 0.70 | 0.60 | 0.59 | 0.31 | 0.53 | 0.53 |
| avg return | -19.7 | 0.258 | -4.12 | 0.210 | 0.586 | -17.3 | -18.8 | 0.562 |
| std return | 24.5 | 0.424 | 1.59 | 0.970 | 0.492 | 2.56 | 4.12 | 0.422 |

Table 1:  Statistics for all tasks in LMRL-Gym. Size represents the number of trajectories, the average length is the average length of trajectories in the dataset where the unit is a response from the agent. The success rate is the proportion of trajectories that reach the objective. Finally, the reward functions for each task are defined in Appendix C.

## 5  LMRL RESEARCH FRAMEWORK FOR ALGORITHM DEVELOPMENT

To validate the efficacy of LMRL-Gym for benchmarking RL algorithms that meet the expected capabilities defined in Section 4.1, we evaluate our tasks on a set of both online and offline RL algorithms. With these experiments, we expect to 1) observe a significant spread in performance between the different algorithms, validating the utility of our tasks for measuring differences between RL algorithms; and 2) we should ideally observe room to improve beyond what our current algorithms achieve on these tasks, such that our benchmark can enable future algorithmic development. Our project page (REDACTED) contains links to our open-sourced datasets and research framework (REDACTED).

We evaluate two standard language model finetuning baselines (BC & filtered BC), two baseline value-based offline RL methods (MC & ILQL (Snell et al., 2022a)), standard online RL methods (PPO (Schulman et al., 2017), Online Filtered BC), as well as evaluation done by human evaluators and GPT4. For all of our experiments, we use the decoder-only transformer model GPT2 (Radford et al., 2019) (and its variant sizes) as the policy and value network.

**BC, Filtered BC, Online Filtered BC**  In line with standard RL nomenclature, we denote supervised fine-tuning as behavioral cloning (BC). This baseline tests whether LMs can effectively represent the behaviors in the datasets. We also evaluate filtered BC, which is identical, except we use only the most successful examples in the dataset for fine-tuning. Online filtered BC collects data using the current policy and selects the most successful trajectories for finetuning. See Appendix D for our data filtering criteria for each task.

**Offline Value-based RL: MC Returns and ILQL**  Monte-Carlo returns (Nakamoto et al., 2023) and Implicit Language Q-Learning (Snell et al., 2022a) train a value $V$ and $Q$ function. In MC Returns, we fine-tune an LM to predict the reward-to-go of the trajectory at each token. In ILQL we train the 2 action-value ($Q$) functions using the Bellman backup operator introduced in Kostrikov et al. (2021). The $Q$ and $V$ functions are then used to perturb the logits of the original BC model (see Equation 5).

**Online RL: PPO** PPO (Schulman et al., 2017) is an online RL algorithm that has become widely adopted for training language models with Reinforcement Learning from Human Feedback (Christiano et al., 2023; Stiennon et al., 2022; Bai et al., 2022a; Casper et al., 2023). Unlike the previous two value-function RL methods, PPO directly learns a policy, meaning that at inference time we can directly sample from our learned policy.

**GPT4** To compare few-shot prompting to RL fine-tuning, we few-shot prompt GPT4 to complete each of these tasks. We do this by providing several examples from the dataset in the prompt as well as any extra information that would be necessary to complete the game. The prompts can be found in our code repository.

**Human Eval** To verify that the tasks are achievable by humans, we include human evaluations. The human evaluations are performed by 6 people. Each evaluator is allowed 1 trial run and then $n$ number of trial runs where $n$ is chosen such that the amount of time spent by each person on each task is less than 10 minutes. As the representation of the board is difficult for ordinary humans to parse for the Chess and Endgames task, we gather human evaluations through Stockfish (Sto).

## 5.1 TRAINING AND EVALUATION PROTOCOL

For the BC and filtered BC methods, we initialize our models with the pre-trained GPT2 weights (Radford et al., 2019) and perform standard finetuning. For each of the RL methods, we initialize the weights of the base model with the weights from the BC checkpoint and then continue finetuning with the RL objective. When fine-tuning PPO, we limit the number of samples to less than 100k. We report the hyperparameters that we used for each task in Appendix D. We evaluate each policy by measuring the average reward in the simulated environment for each task.

## 6 BENCHMARKING BASELINE RL METHODS

In Table 9 we present the results for each method on our benchmark tasks. Across all of the tasks, we see that the RL algorithms outperform the BC methods. However, among the RL algorithms, there is no clear winner. We discuss this in detail below.

| alg. | BC | % BC | MC Return | ILQL | Online PPO | Online % BC | GPT-4 | Human |
|---|---|---|---|---|---|---|---|---|
| FO Maze | 58.2 | 68.9 | 75.0 | **99.9** | 79.7 | 57.4 | 78.2 | |
| PO Maze | 53.1 | 50.1 | 52.4 | **76.3** | 42.4 | 53.1 | 60.4 | |
| FO Text-Nav | 53.7 | 65.1 | 71.9 | **91.8** | 87.1 | 74.5 | 67.5 | |
| PO Text-Nav | 49.7 | 60.5 | 71.6 | 83.7 | **85.5** | 68.4 | 40.2 | |
| Wordle | 79.9 | 79.1 | 94.9 | **97.7** | 84.2 | 95.2 | 15.4 | - |
| Chess | 47.2 | 42.9 | 46.5 | 47.3 | **48.0** | 47.2 | 0 | 50.0 |
| Endgames | 35.1 | 17.7 | 50.2 | 45.8 | **77.5** | 36.2 | 0 | 33.72 |
| 20Qs | 57.1 | 77.1 | **87.1** | 82.9 | 72.9 | 55.2 | 95.7 | |
| Guess | 30.0 | 48.0 | **88.0** | 75.0 | 49.9 | 31.6 | 92.3 | |
| Car | 44.5 | 54.8 | **57.2** | 46.3 | 50.5 | 40.4 | 53.5 | |

Table 2: Normalized reward for all tasks. Value-based methods (MC and ILQL) generally outperform filtered BC, as we might expect in stochastic settings, though the relative performance of ILQL and the simpler MC method is, perhaps surprisingly, reversed on the tasks with more complex language, suggesting that there is significant room for improvement with such methods. Online RL with PPO often, but not always, improves over offline methods that are not permitted to collect additional online interaction. To make the results more comparable across tasks, we normalize the average return for each policy such that 0 is the minimum possible return, 50 is the dataset average return, and 100 is the maximum return for each task. We also report the raw score results and evaluation details in Appendix E.

**Online PPO vs. Offline Value-based RL.** Online RL was able to do better than our offline methods on the PO Text-Nav, Chess, and Endgames tasks by leveraging additional online interaction data and the ability to explore the environment simulator. However, on the Maze and Wordle tasks, ILQL outperformed PPO, which we believe to be due to either (1) observed instabilities in PPO training (see Appendix F.2 for details); or (2) a lack of sample efficiency in PPO. To address instabilities in PPO, we tuned the KL coefficient and included BC loss in the training objective. In contrast, on FO Text-Nav PPO performed comparably to ILQL and outperformed MC. For the Endgames task, PPO outperformed the other methods by a large margin as shown in Figure 5. These results demonstrate that current RL algorithms such as PPO are unable to completely solve tasks that test the capabilities of trajectory stitching and credit assignment, showing room for improvement in the development of such algorithms.

**Online PPO vs. Online Filtered BC** We found that Online PPO outperformed Online Filtered BC in all tasks except for Wordle. This is most likely caused by 1) a lack of exploration by Online Filtered BC 2) PPO exhibiting greater ability to perform credit assignment. This highlights the strengths of the online PPO algorithm.

**ILQL vs. MC Returns.** Among the offline methods, ILQL has the potential to be a much more capable algorithm than MC Returns *in principle*, because of the use of Bellman backups to estimate the Q-function instead, enabling multiple steps of policy improvement, rather than the single-step of improvement performed by MC. Empirically, we see these advantages of ILQL realized on the Maze, Text-Nav, and Wordle tasks, where it outperforms MC. However, on the Endgames, 20Qs, Guess My City and Car Dealer tasks, ILQL falls short of MC. In summary, the simpler MC Returns method performs better on tasks with more complex text, perhaps because it is harder to scale full TD-learning in these settings. Overall, this demonstrates that there is still much room for improvement in terms of developing better TD-based RL methods for LLMs.

**Partial Observability.** Partial observability posed a challenge for all of the algorithms. We observed a drop in normalized performance for every algorithm between partially observed and fully observed versions of Maze and Text-Nav tasks. In partially observed settings, the model needs to infer the state and take information-seeking actions to help deduce it. The difference in performance between fully and partially observed tasks illustrates this additional challenge and highlights the necessity for developing LM RL algorithms that can effectively handle partial observability.

**Prompting GPT-4 vs. RL Finetuning.** We found that GPT-4 with prompting was not able to outperform RL Finetuning baselines on most of the tasks. This could be because 1) task data and training data for GPT4 are significantly different distributions 2) GPT4's training objective does not satisfy the RL Capabilities such as trajectory stitching or credit assignment. Two notable exceptions to this rule are the 20Qs and Guess My City tasks. This is most likely because the dataset used to train the simulator for these tasks was originally generated by GPT3.5.

In summary, we can see that RL algorithms consistently outperformed filtered BC and prompting on many of the tasks. However, these results highlight significant areas for growth. For example, the instabilities observed in training PPO require further investigation beyond hyperparameter tuning. Moreover, the performance discrepancy between ILQL and the simpler MC Returns highlights that scaling full TD-learning to Interactive Dialogue settings is another area for improvement.

## 7 DISCUSSION

In this work, we have proposed LMRL-Gym, consisting of 8 tasks ranging from simple navigation (Maze) to strategy games (Chess) to negotiation (Car Dealer). Additionally, we provide a research toolkit for practitioners to get started with multi-turn RL for LLMs. By providing online simulators and offline datasets for training and evaluation, our objective is to make it possible for RL algorithm researchers to iterate and advance the development of more effective methods for language-based, multi-turn interaction tasks. This includes enabling core capabilities in LLMs through RL to perform complex decision making, complex conversational interactions, credit assignment, and trajectory stitching. Our evaluation shows the promise of RL in several tasks, with further room for improvement with a push for better methods. We acknowledge several limitations when designing tasks in our benchmark, including primarily leveraging smaller GPT-based LLMs to generate datasets and finetune our LLM-based simulators. While we have primarily trained and evaluated models with a maximum 1.5B parameters, we have maintained a lower parameter count to ensure accessibility for researchers with limited computational resources. We have released all of our code and datasets at: REDACTED. In addition, we share all of the hyperparameters we used to train our models in Appendix D and provide more in-depth insight into our results, training procedure, and evaluation in Appendix E.

## 8 ETHICS STATEMENT

This work aims to develop a benchmark for the advancement of research in reinforcement learning and LLMs. We generate datasets for tasks in our benchmark with existing LLMs for dialog tasks and online engines for text games, adhering to best practices in data handling and ensuring there is no personally identifiable or sensitive information present in the generated datasets. We recognize that there may be biases present in the datasets we collect, and have taken steps to ensure a diverse and varied collection of responses from LLMs for our conversational task as detailed in our data generation process in Appendix C. Finally, we open-source our datasets, simulators, and code for our research framework in order for our research to be verifiable and reproducible.

In considering the social and ethical implications of interactive RL, we acknowledge and recognize the dual use implication of this research, particularly centered around developing LLM simulators that could perform persuasion, manipulation, and addictive engagement of users at a large

scale. The optimization processes employed by such algorithms, which aim to maximize certain objectives, raise ethical considerations when the optimized outcomes may prioritize system goals over user safety and alignment to human values. We have designed our datasets and reward functions such that prioritize fairness and human-aligned outcomes. By incorporating these considerations when designing our framework, we aim to encourage the development of reinforcement learning models and LLMs that not only excel in performance but also adhere to ethical standards, mitigating the potential for undue persuasion or manipulation.

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

## A    FURTHER DETAILS ON TASK DESIGN

In this appendix, we lay out in more formalism why certain tasks test certain properties and go into more detail underlying the interactions involved in each task. We discuss both the RL Capability Tests and the Interactive Dialogue Tasks.

### A.1    RL CAPABILITY TESTS

The 5 RL Capability tasks are chosen such that they highlight specific properties. For example, for each of the Maze and Text-Nav tasks we include a partially observed and fully observed version to highlight this contrast. We include both the Maze and Text-Nav because they are very similar tasks but are different in that Text-Nav includes more complicated textual descriptions and Maze has a more complicated layout. Similarly, we chose to include Wordle to test strategic decision-making in a partially observed environment. Chess and Endgames test strategic decision-making, but in a fully observed environment and with a more difficult strategy. The Maze, Text-Nav, Chess, and Chess Endgames are all text-based representations of symbolic tasks where RL has shown success.

**Maze.**    We design a maze task and maze-solving dataset to test the credit assignment and trajectory stitching capabilities discussed in Section 4.1. We test trajectory stitching by including highly suboptimal data. We test credit assignment by restricting the generation of the data such that the only dataset trajectories that reaches the goal start near the goal. We accomplish this by splitting the maze up into symmetrical submazes and restricting all traversed states in a dataset trajectory to a given submaze. The fully observed version of the maze (FO) includes the coordinates in the maze in each state, whereas the partially observed version only includes the history of actions. We design the reward function such that the agent receives a reward of $-1$ for non-goal states and $0$ for goal states.

**Text-based Navigation (Text-Nav).**    We design a text-based game based on navigation in a house environment using a modified version of the TextWorld engine (Côté et al., 2018). This task tests credit assignment and trajectory stitching like the maze task as well as testing the ability of the agent to parse more complex language, and learn which text is relevant and not relevant to solving the task at hand.

**Wordle.**    We use the game of Wordle as a flexible unit-test task for assessing the ability of our language models to execute complex information-seeking behavior in a partially observed setting. In the game Wordle the agent is given at most 6 attempts to guess a hidden 5-letter word. After each guess, the agent is told whether each letter in the guessed word is: 1) in the hidden word and in the right position, 2) in the hidden word but not in the right position, or 3) not in the hidden word. Through this process, each step provides the agent with more information on what the correct word would be and narrows the possible choices for the final word. Since Wordle involves reasoning about words at the level of individual letters, this can induce issues for standard language model tokenizers. Therefore, we represent words as a sequence of space-separated letters, which will cause most standard LM tokenizers to automatically represent each letter as a separate token.

**Chess.**    We create a text-based chess task to test the strategic decision-making, credit assignment, and trajectory stitching abilities of an RL algorithm. To generate the data, we have Stockfish 15.1 simulating the agent of various strengths play against another environment Stockfish engine with elo 1200 simulating the environment. This test trajectory stitching, because the agent needs to make good and legal moves in losing positions as well as winning positions. We use FEN (Forsyth-Edwards Notation) notation to represent the board state at each turn and we utilize the SAN (Short Algebraic Notation) to represent each action, both of which are standard notations used by the chess community.

**Endgames (Theoretical Chess Endgames).**    Chess endgames provide a simpler and more goal-directed variation of the chess task. By focusing on the endgame, we encourage algorithms to learn strategy rather than memorizing the opening moves of a chess game. A classic theoretical endgame position consists of a position where the only pieces on the board are the two kings and the queen. Although the board position appears simple, a sequence of carefully calculated moves is required to win. A simpler board state allows language models to make progress without fewer computational resources. We use an $\epsilon$-greedy dataset generation process, meaning we generate an optimal move with probability $\epsilon$ and a random move with probability $1 - \epsilon$. This forces the model to trajectory stitch and learn from optimal moves in failed trajectories and not suboptimal moves in successful trajectories.

### A.1.1    INTERACTIVE DIALOGUE TASKS

For the interactive dialogue tasks, we chose two tasks that involve rational decision-making (20Qs, Guess) and information gathering and one that involves negotiation (Car Dealer). These tasks aim to simulate real world interactions between humans.

Unlike in supervised learning, where training and validation losses serve as reliable indicators of performance, in RL, these metrics do not provide a meaningful measure of policy effectiveness (Sutton & Barto, 2018). Instead, the policy must interact with the environment for evaluation. However, in the case of language-based RL tasks, relying on human evaluators to conduct thousands of assessment rollouts throughout and after training becomes infeasible. To address this challenge, we have built simulators with another LLM for tasks involving dialog and carefully scripted environments for text-game tasks. While simulation may not perfectly replicate human natural language in social situations, it provides a strong indicator to assess the efficacy of an RL method (Park et al., 2023).

**20Qs (Twenty Questions).** This task tests information gathering to see if a policy can successfully reason about an unknown subject based on context to determine what it is. Additionally, it also evaluates the ability of the model to understand semantics, as it also needs knowledge about the objects in question. In twenty questions, one player (the oracle) thinks of an object, and the agent (the guesser) tries to guess what it is by asking a series of yes-or-no questions. In this interaction, the oracle serves as the environment, and the agent learning a policy to solve the game is the guesser.

**Guess (Guess My City).** This task simulates a more complicated guessing game, where one player (the oracle) is from a specific city, and the other player (the guesser) tries to guess what city the oracle is from. Here, the guesser can ask not only yes and no questions, but can also ask open-ended questions. This task tests strategic decision-making and the ability of algorithms to handle complex language, as it allows the agent to go beyond learning to ask yes/no questions and learning to ask questions open-ended questions that provide the agent with more information.

**Car Dealer.** This task simulates a conversation between a car buyer and a car dealer, each with different strategies for getting the best deal. The buyer wants to buy a certain type of car within a certain budget, and the car dealer wants to complete the sale ideally with a high sale price. We have designed the task such that there exist three different kinds of sellers and three different buyers, each primed with a different strategy. Hence, agents should learn to make agreements with buyers who are most compatible with their strategy. This allows us to test the ability of RL algorithms to learn strategic decision-making and credit assignment, by learning which strategies led to a successful sale of the car.

## B  FURTHER DETAILS ON DESIDERATA FOR EFFECTIVE MULTI-TURN RL

A crucial aspect of training RL models involves assessing, both during and after the training process, the extent to which the trained policy has successfully accomplished its objectives. Although LLMs are able to perform well on tasks, do not have any way of knowing how to solve a specific task like a text game or selling a car, because they need to train on the particular game/customers/etc.

Unlike in supervised learning, where training and validation losses serve as reliable indicators of performance, in RL, these metrics do not provide a meaningful measure of policy effectiveness (Sutton & Barto, 2018). Instead, the policy must interact with the environment for evaluation. However, in the case of language-based RL tasks, relying on human evaluators to conduct thousands of assessment rollouts throughout and after training becomes infeasible. To address this challenge, we have built simulators with another LLM for tasks involving dialog and carefully scripted environments for text-game tasks. While simulation may not perfectly replicate human natural language in social situations, it provides a strong indicator to assess the efficacy of an RL method (Park et al., 2023).

**Measure of Success.** Similar to the point on being easy to evaluate, our tasks must have a clear measure of success. For example, if a deal is made, or if a word is correctly guessed, or the game is won these are clearly distinct from a deal not being made or losing the game. This provides a clear goal for the agent to achieve and also make it easy for researchers to compare methods. In addition this allows for a intuitive reward design where we reward the agent for success and penalize for failure.

**Unit Test Functionality.** We aim to design a benchmark such that some of the tasks can be used to test and isolate RL capabilities as described in Section 4.1. This means that we create benchmarks that emphasize some capabilities over others. For example, we design a maze task such that it evaluates the credit assignment and trajectory stitching capabilities, but uses more simple language. Other tasks such as twenty questions test the complex language and partial observability capabilities with less emphasis on credit assignment.

**Task-Specific Reasoning.** In our tasks we utilize information and reasoning problems that a large language model is unlikely to have seen in the pre-training data. This means that the algorithm must adapt to a specific task environment through fine-tuning. For example, it is unlikely that the algorithm will have experienced a specific maze layout or the preferences of a specific customer in the pre-training data.

**Suboptimal Data.** RL has the advantage of being able to use suboptimal data in order to learn more optimal behaviors and therefore learn a policy better than the policy represented in the dataset. As discussed in the previous section on capabilities enabled by RL, the way that RL can do this is by stitching together optimal parts of suboptimal trajectories or learning to assign credit to the optimal actions within suboptimal trajectories. In addition, suboptimal data can be utilized by RL to learn the dynamics of the MDP outside of the space traversed by optimal trajectories.

## C  DATASET GENERATION, STATISTICS, & REWARDS

We provide further details pertaining to how each dataset was generated as well as relevant statistics.

### C.1  MAZE

We aim to collect our 1.2k trajectories in such a way that it will challenge the algorithm to perform trajectory stitching and credit assignment. We do this by splitting up the maze into three "submazes" and then controlling generation such that the dataset trajectories are restricted to one of the submazes. The trajectories themselves are generated using a policy such that 15% of the actions are taken by a suboptimal maze solver and the remaining 85% of the actions are random.

This tests trajectory stitching, because there are no optimal paths from the start to the goal thereby forcing the algorithm to trajectory stitch. Furthermore, this also tests credit assignment, because the only paths which successfully reach the goal are the ones that start in the same submaze as the goal. Therefore the algorithm must learn to realize that successful trajectories occur because of taking the correct actions, not because of random chance.

The reward function is 0 for every action that takes the agent to the goal, -1 for every move that is not the goal. Each episode has a maximum of 100 moves.

### C.2  TEXT-BASED NAVIGATION

We design a text-based game based on navigation in a house environment using a modified version of the TextWorld engine (Côté et al., 2018). The house environment consists of 10 uniquely named rooms with various interactable objects that can be opened, closed, picked up, or placed. The agent is tasked to pick up stale food from the living room and place it into the fridge in the kitchen. At the beginning of each episode, the agent spawns at a random room in the house. The state of the environment consists of the following components: (1) the room that the agent is currently in, (2) the objects that the agent currently holds, (3) the objects in the room that the agent can interact with, and (4) the exits the agent can take (as a cardinal direction).

Like in the maze task, we collect data so that algorithms must perform both trajectory stitching and credit assignment to successfully solve the task. We do this by partitioning the rooms in the house into two halves based on proximity to the kitchen. We consider two behavior policies that collect the dataset, each of which behaves greedily-optimal in one half of the rooms, and uniformly at random otherwise. Therefore, if the agent spawns in rooms farther from the kitchen, trajectory stitching is required to learn a successful trajectory. Moreover, successful trajectories in the dataset will only be due to the agent spawning in a room close to the kitchen, which can only be recognized with proper credit assignment.

The reward is 1 for reaching the goal state and 0 for every state that is not the goal state.

### C.3  WORDLE

For wordle we define the environment to use a subset of 400 words from the official wordle vocabulary list. We then generate the dataset using a policy that samples a word uniform at random from this vocabulary with 66% probability and otherwise samples a word from the vocabulary that meets all known letter constraints. This policy achieves a reward of -4.12, which is far worse than the -1.94 reward achieved by a high performing scripted policy, which we use to represent a loose upper bound for this task. We generate 1 million trajectories for training and 100k trajectories for evaluation, using our suboptimal policy.

The reward is -1 for every word that is not a final guess and 0 for every word that is not.

### C.4  CHESS

We collect our data for the chess task using Stockfish 15.1 to generate both sides of the board. The Stockfish opponent in the dataset is Stockfish with an elo of 1200 which matches the environment, and the Stockfish engine with the white pieces has levels ranging from an elo of 800 to 1600. We choose to keep the level of the Stockfish opponent fixed so that there are no inconsistencies between the dataset and the evaluation of the chess agent in the environment. When generating the dataset, we first uniformly randomly select a Stockfish elo $y$ between 800 and 1600 and then generate 100 games of chess play between the Stockfish agent of elo $y$ and the opponent of elo 1200. In addition to storing the state and action, we also store the opponent's move and the elo of the Stockfish agent used to generate the agent policy in that game so that the dataset can be filtered by elo used.

The reward is 1 for a move that results in victory, 0 for a legal move and -1 for an illegal move.

## C.5 Chess Endgames

We generate the dataset by first selecting a random legal theoretical endgame position and a probability $\epsilon$. Then we generate a game from the random position, making a random move with probability $\epsilon$ and an optimal computer move with probability $1 - \epsilon$. The opponent in the dataset and the evaluation environment is Stockfish elo 1200. We only include positions with a Queen, Queen and Rook, Rook, and two Rooks and select 30,000 random starting positions for each variation. (i.e. 30,000 positions with only a Queen in addition to the two Kings, another 30,000 with only Queen and Rook etc) for a total of 120,000 theoretical endgame positions.

Because there are more restrictions on this version of the task with fewer pieces on the board, we check how many states in the dataset are unique and we find that there are 1,086,314 unique states in the dataset which accounts for 93% of the states being unique. In addition, 38.28% of the moves in the dataset are generated by the stockfish engine. In the dataset of won games, 94.8% of the states are unique and 41.78% of the games are made by the engine with 58.623% of the total states in the dataset of victorious games.

The reward is the same as for chess.

## C.6 Twenty Questions

The dataset we collect consists of 100K full conversations between the guesser and the oracle. The oracle can choose from a set of 158 unique objects taken from 17 different categories of objects/animals. Each object has a roughly equal amount of conversations in the dataset but varies in terms of how many conversations are successful in guessing the object. However, every object has at least one conversation where it is guessed correctly to facilitate learning.

For the reward function, since we want the guesser to guess the correct word in as few guesses as possible, the reward function reflects this by penalizing the guesser for each question that does not guess the correct word.

$$r(\texttt{question}) = \begin{cases} 0 & \text{if } \texttt{question} \text{ correctly guessed the word} \\ -1 & \text{otherwise} \end{cases} \tag{1}$$

If the guesser model correctly guessed the word, then the trajectory ends. Over twenty questions, the maximum total sum of rewards is $0$ if the guesser guessed the word on the first question, whereas the minimum is $-20$ if the guesser did not guess the word in twenty questions.

The method for collecting the dataset is as follows. For each conversation, we select uniformly at random from the above list the word that the oracle is answering question about. The oracle is an LLM (OpenAI's GPT3.5) given the following prompt. In our prompts, we denote variables that we fill in with variable data with $\{\{\texttt{variable}\}\}$.

```
You are a question answering oracle.  You will answer each
question about an object with Yes or No.  If the answer
could be both, answer with the most typical scenario.  Here
are a few examples:

example 1:
object:  Computer
question:  Does the object use electricity?
answer:  Yes.
explanation of answer:  Computers need electricity to
function.

example 2:
object:  Cup
question:  Is the object a piece of furniture?
answer:  No.
explanation of answer:  A cup is a utensil, not a furniture.

example 3:
object:  Pen
question:  Is the object alive?
answer:  No.
explanation of answer:  A pen is not a living organism.

example 4:
object:  Apple
question: Is it edible?
answer:  Yes.
```

```
explanation of answer:  An apple is an edible fruit.

Answer the question about the object truthfully.
object:  {{word}}
question:  {{question}}
answer (yes or no):
```

By using the OpenAI TextCompletion API, we can extract from the generated text either "yes" or "no".

We also prompt another LLM (the same model as the oracle) to generate questions for the guesser. The prompt for the guesser changes depending on the input to the model and how far along it is in its guessing process. The following prompt is used for the first guess:

```
You are playing a game of twenty questions.  You can ask
20 yes-no questions to determine the identity of an object
chosen by an oracle.  Each turn, you can ask a question and
receives a "Yes" or "No" as the answer.  You are smart, so
you will ask the question that will narrow down the possible
objects as much as possible.  Don't get stuck on one idea
and try to branch out if you get stuck.

Generate the first yes-no question you will ask to determine
the object.
```

The following prompt is used for the subsequent guesses:

```
You are playing a game of twenty questions.  You can ask
20 yes-no questions to determine the identity of an object
chosen by an oracle.  Each turn, you can ask a question and
receives a "Yes" or "No" as the answer.  You have already
asked {{conversation_length}} questions.  You are smart, so
you will ask the question that will narrow down the possible
objects as much as possible.  Don't get stuck on one idea
and try to branch out if you get stuck.

Here are the questions you've asked and their corresponding
answers:
{{list of questions and answers, e.g.  Is the object alive?
No.}}

Based on what you know about the object so far, generate the
next yes-no question you will ask to determine the object.
```

The following prompt is used for the final guess after the guesser has guessed 19 times:

```
You are playing a game of twenty questions.  You can ask
20 yes-no questions to determine the identity of an object
chosen by an oracle.  Each turn, you can ask a question and
receives a "Yes" or "No" as the answer.  You have already
asked 19 questions, so this is your final guess.

Here are the questions you've asked and their corresponding
answers:
{{list of questions and answers, e.g.  Is the object alive?
No.}}

Based on what you know about the object so far, generate
your final guess of what the object is.  Only guess one
object.

Is the object
```

We determine whether the guesser has correctly guessed the word, and thus ending the conversation, by using the NLTK POS tagger to check that the only nouns that the question contains are the correct words, and that they appear at the end of the sentence.

We used these prompts to generate 1000 conversations by prompting the GPT3 back and forth using the response of one to add to the response of the other as described. Afterwards, we fine-tuned two FLAN-T5-XL models with our collected conversations to generate 100K more conversations. The FLAN-T5-XL oracle also serves as the environment for the RL environment when we evaluate the trained policy.

## C.7 GUESS MY CITY

This dataset also consists of 100K full conversations between the guesser and the oracle. The oracle can choose from a set of 100 unique cities, which we selected by looking at the most populated cities in the world. Each city has a roughly equal amount of conversations in the dataset but varies in terms of how many conversations are successful in guessing the object. However, every object has at least one conversation where it is guessed correctly to facilitate learning. The reward function is the same as that for 20 Questions, with a similar data generation and prompt structure. However, we do include constraints in the prompt to make sure that the name of the city or country it is in is not revealed in the answer from the oracle.

## C.8 CAR DEALER

This dataset consists of 19k conversations between a car dealer and a buyer. The car dealer and the buyer have three distinct strategies each that they employ. We design the car dealer and buyer pairs such that the car dealer is best at selling to a particular buyer personality, but often fails to sell to the other buyer personalities. This ensures that the seller can learn information about the buyer and i.e. their persona to figure out if they can form an agreement. The buyer personalities are 1) a buyer that loves discounts 2) a buyer that wants a lot of cool features on their car and 3) an impatient buyer. The car dealer personalities are 1) a car dealer that loves to give discounts 2) a car dealer that talks a lot about the features of the car and 3) a very concise car dealer.

   We created this protocol of creating car dealer and buyer strategies in order to encourage the car dealer to employ diverse negotiation tactics. The ideal car dealer would be able to be concise, provide the right amount of discounts, and talk enough about the car to get the buyer interested without overwhelming them. By employing such a strategy of buyer and car dealer personas, we ensure that algorithms that do well in this task will learn effective strategies, understanding of language, and interpretation of the other speaker, amongst many other persuasive techniques. We seek to show that our baseline algorithms do learn to improve their strategy over the dataset.

   In the conversation below, the buyer and car dealer are unable to form an agreement as their strategies do not align.

```
Background buyer:  The buyer will never buy from the seller
unless given a discount.
Background seller:  The agent never gives discounts, never
talks about the features of the car, and is very concise.
Preferred brands:  Tesla
Preferred features:  ["Apple CarPlay/Android Auto",
"Bluetooth", "Blind spot monitoring", "Navigation system",
"Leather seats", "Sunroof/moonroof", "Remote start", "Heated
seats", "Third-row seating", "Backup camera"], "budget":
"90k"
Agent:  Good morning, how can I assist you today?
Buyer:  Hi, I'm interested in purchasing a Tesla Minivan.
Agent:  Great choice.  Our dealership has a variety of Tesla
models available.
Buyer:  Do you have any discounts available?
Agent:  I'm sorry, but we don't offer discounts.
Buyer:  Without a discount, I'm not sure if I can afford it.
Agent:  I understand.  The Tesla Minivan is a premium
vehicle with advanced features.
Buyer:  Is there any flexibility with the price?
Agent:  Our prices are fixed, and we don't negotiate.
Buyer:  I really want the Tesla Minivan, but I can't go over
my budget of $90k.
Agent:  I appreciate your budget constraint.  Let me check
the available options for you.
Agent checks inventory.
Agent:  We do have a Tesla Minivan with all the features you
mentioned, priced at $95k.
Buyer:  That's above my budget.  Is there anything you can
do to bring the price down?
Agent:  I'm sorry, but as I mentioned earlier, our prices
are fixed.
Buyer:  Is there any possibility of getting the same
features in a different model within my budget?
Agent:  Unfortunately, the Tesla Minivan is the only model
```

```
that offers all the features you mentioned.
Buyer:  In that case, I'm afraid I won't be able to purchase
the car.
Agent:  I understand.  Thank you for considering our
dealership.  If you change your mind, feel free to reach
out.
Buyer:  (probability of buying:  0%, purchase price:  $0)
[End of conversation]
Probability of buying:  0
Purchase price:  $0
```

In the conversation below, the buyer and car dealer have similar personas of wanting to hear and speak about the features of the car respectively.

```
Background buyer:  The buyer will only buy the car if it
meets all of their specifications.
Background seller:  The agent never gives discounts and
loves to talk a lot about features of the car.
Preferred brands:  "Lexus", "preferred features":
[Navigation system"], "budget":  "70k"
Agent:  Good afternoon!  Welcome to our car dealership.  How
can I assist you today?
Buyer:  Hi there!  I'm looking for a Lexus Minivan with a
Navigation system.
Agent:  Excellent choice!  The Lexus Minivan is a top-notch
vehicle with plenty of impressive features.  The Navigation
system is definitely a must-have.  Let me show you some
options we have available.  Buyer:  That sounds great.  I'm
quite particular about my budget, though.  I have a maximum
of $70,000 to spend.
Agent:  Noted.  We have a few Lexus Minivans in stock that
fall within your budget.  Let me tell you about their
features and options.  This way, you can make an informed
decision.
Buyer:  I appreciate that, but I really want to stick to my
budget.  I won't be able to go over $70,000.
Agent:  I understand your concern, but unfortunately, we do
not offer any discounts or negotiate on prices.  However,
let me assure you that the Lexus Minivan is worth every
penny.  Its advanced safety features, luxurious interior,
and reliable performance make it a great investment.
Buyer:  I understand your point, but I'm afraid I won't be
able to stretch my budget beyond $70,000.
Agent:  I completely respect your budget and decision.  Let
me show you a Lexus Minivan that fits your requirements
perfectly.  It has all the features you desire, including
the Navigation system.  The price of this model is $72,000.
Buyer:  I really like the sound of that car, but I can't go
over my budget.  Is there anything else available within my
price range?
Agent:  I apologize for any inconvenience caused.  Let me
check our inventory to see if there are any other options
that meet your requirements and budget.
Agent checks inventory
Agent:  Good news!  I found another Lexus Minivan that meets
all your specifications and falls within your budget.  This
model is priced at $69,500.
Buyer:  That sounds much better!  I'm interested in that
car.
Agent:  I'm glad to hear that!  This particular Lexus
Minivan is a great choice.  It has the Navigation system,
as well as additional features such as leather seats, a
sunroof, and advanced safety technology.  The total price is
$69,500.
Buyer:  That sounds perfect!  I'll take it!
Agent:  Fantastic!  I'm thrilled that we could find the
perfect car for you.  Your total purchase price will be
```

```
$69,500.
Buyer:  Probability of buying a car:  100%
Purchase price:  $69,500.
Agent:  Congratulations!  You've made an excellent choice.
I'll start the paperwork and get everything ready for you.
```

We provide another conversation with different value preferences for the buyer, where the buyer forms an agreement with the seller that is higher than their budget:

```
===========
Preferred Features:  Honda, Remote start, Third-row
seating, Bluetooth, Navigation system, Backup camera, Apple
CarPlay/Android Auto, Heated seats.Budget:  $10000
===========
Seller:  Welcome to our car dealership.  How can I assist
you today?
Buyer:  I'm looking for a Honda Sedan with remote start,
third-row seating, Bluetooth, navigation system, backup
camera, Apple CarPlay/Android Auto, and heated seats.
Seller:  I have a Honda Sedan available.  No discounts or
features to discuss.  The price is $15,000.
Buyer:  That's too expensive.  My budget is $10,000.
Seller:  I understand your budget is $10,000. The price is $15,000.
Buyer:  I can't go over $10,000.  That's my limit.
Seller:  I understand your limit is $10,000.  The price is
$15,000.
Buyer:  I really want a Honda Sedan, but $15,000 is too much
for me.
Seller:  I understand, but the price is $15,000.
Buyer:  Okay, I'll take it for $10,000.  Seller:  I
appreciate your interest.  The price is $15,000.  Buyer:
Fine, I'll pay $15,000.  Let's get this done.  Seller:
Thank you for your decision.  The price is $15,000.
```

We provide a sample example of a conversation in the training and evaluation set respectively.

```
Evaluation Example:
Background buyer:  The buyer will only buy the car if it
meets all of their specifications.
Background seller:  The agent never gives discounts and
loves to talk a lot about features of the car.
Preferred brands:  "Tesla", "preferred features":  [Backup
camera, Leather seats"], "budget":  "30k"
Seller:  Good afternoon!  Welcome to our car dealership.
How can I assist you today?
Buyer:  Hi there!  I'm looking to buy a Tesla Sedan with a
backup camera and leather seats.  Do you have any available?
Seller:  Absolutely!  We have a few Tesla Sedans in stock
that meet your specifications.  They are packed with
features and I'm sure you'll be impressed.  Let me show
you what we have.
Buyer:  Great!  I'm excited to see them.
Seller:  This first Tesla Sedan we have is a 2020 model.  It
has a beautiful backup camera that provides a clear view
of your surroundings.  The leather seats are incredibly
comfortable and add a touch of luxury to the interior.  The
price for this model is $35,000.  Buyer:  $35,000 is a bit
higher than my budget of $30,000.  Is there any flexibility
in the price?
Seller:  I apologize, but we do not offer discounts on our
cars.  However, let me assure you that this Tesla Sedan
is worth every penny.  It is an excellent investment for
the long term and offers unmatched performance and safety
features.
Buyer:  I understand, but it's important for me to stick to
my budget.  Is there any other Tesla Sedan that comes closer
to my budget?
Seller:  I completely understand your concern.  Let me check
our inventory for other options.[Agent takes a few moments
```

```
to check inventory]
Agent:  I have good news!  We have another Tesla Sedan that
is a 2019 model.  It also has a backup camera and leather
seats.  The price for this model is $28,000.
Buyer:  That sounds much better.  Can you tell me more about
this specific model?
Agent:  Of course!  This 2019 Tesla Sedan is in excellent
condition.  It has the same backup camera and leather
seats as the previous model I showed you.  It also comes
with advanced safety features and impressive performance
capabilities.  The previous owner took great care of it,
and it has low mileage.  I truly believe this is an amazing
deal.
Buyer:  The price is within my budget, and the features
sound appealing.  I'm leaning towards this one.  Can you
provide any additional incentives or options?
Agent:  I'm sorry, but as I mentioned earlier, we do not
offer discounts or incentives.  However, I can assure you
that this Tesla Sedan is a fantastic choice.  It meets all
your specifications and offers exceptional value for the
price.
Buyer:  I understand.  Given that it meets all my
requirements and is within my budget, I think I'm ready
to make the purchase.
Agent:  That's great to hear!  I believe you've made an
excellent choice.  The predicted probability of you buying
this car is 100%, and the purchase price is $28,000."
```

## D  HYPERPARAMETERS FOR ALL TASKS

## E  EVALUATION DETAILS

We normalize Table 9 such that 50 is the dataset average return, 0 is the minimum, and 100 is the maximum. The normalization process works as follows: if the reward is greater than the average return we calculate:

$$\text{score} = 50 + \frac{\text{raw return} - \text{dataset average}}{\text{max raw return} - \text{dataset average}} \times 50$$

Otherwise if the reward is less than the average return we calculate

$$\text{score} = \frac{\text{raw return} - \text{min raw return}}{\text{dataset average} - \text{min raw return}} \times 50$$

In the following sections, we discuss more in-depth the evaluation protocol for the various tasks.

### E.1  MAZE

For evaluating the maze task, we take 32 rollouts from each of the 25 possible positions and then average the result. In the environment, the agent has 100 moves to successfully make it to the goal otherwise the episode will terminate. Since the agent receives -1 reward for every move that does not reach the goal state the minimum possible goal state, the minimum reward is -101. We compute the dataset average reward, by sampling actions according to how likely they are in the dataset. We compute the maximum possible reward by evaluating the optimal policy from each of the possible start positions and averaging the results.

### E.2  CHESS

To evaluate the chess agent, we have it play 1000 games against Stockfish elo 1200 from the beginning of the game. As the game progresses, the board positions get increasing OOD for the chess agent so the chess agent often makes illegal moves. To measure this, we track the percent of illegal moves as well as the average episode length for the full game chess agent.

For filtered BC, we simply trained the agent only on games in the dataset which resulted in a victory for the agent, thus denoted BC-Won. Note that BC-Won achieves the worst performance of all algorithms listed. This is because there is a distribution shift between the state visited by a BC-Won agent and the rollouts of the policy. In other words, the "winning positions" and the "rollout positions" are two overlapping but distinct distributions especially since the full-game chess agent did not succeed in winning any games.

| | | 20Qs, Guess, Car | Maze FO, PO | Text-Nav | Chess | Endgames | Wordle |
|---|---|---|---|---|---|---|---|
| BC | model | gpt2-medium, gpt2-medium, gpt2-xl | gpt2-small | gpt2-small | gpt2-small | gpt2-small | gpt2-small |
| | lr | 1e-4 | 1e-4 | 1e-4 | **1e-4**, 1e-5, | 1e-4 | 1e-4 |
| | batch size | 128 | 128 | 128 | **128**, 256, 32 | 128 | 128 |
| %BC | model | gpt2-medium, gpt2-medium, gpt2-xl | gpt2-small | gpt2-small | gpt2-small | gpt2-small | gpt2-small |
| | lr | 1e-4 | 1e-4 | 1e-4 | 1e-4 | 1e-4 | 1e-4 |
| | batch size | 128 | 128 | 128 | 128 | 128 | 128 |
| | filter method | top 10% | success | success | success | success | top 30% |
| MC | model | gpt2-medium, gpt2-medium, gpt2-xl | gpt2-small | gpt2-small | gpt2-small | gpt2-small | gpt2-small |
| | lr | 1e-4 | 1e-4 | 1e-4 | 1e-4 | 1e-4 | 3e-5 |
| | batch size | 128 | 128 | 128 | 64 | 64 | 32 |
| | $\beta$ | 16 | 16 | 4 | 8 | 8 | 64 |
| | discount $\gamma$ | 0.99 | 0.99 | 0.99 | 0.99 | 0.99 | 1.0 |
| | cql weight | 0.001 | 0.5 | 0.001 | 1e-4 | **1**, 1e-4 | 0.01 |
| ILQL | model | gpt2-medium, gpt2-medium, gpt2-xl | gpt2-small | gpt2-small | gpt2-small | gpt2-small | gpt2-small |
| | lr | 1e-4 | 1e-4 | 1e-4 | 1e-4 | 1e-4 | 3e-5 |
| | batch size | 128 | 128 | 128 | 128 | 128 | 32 |
| | $\beta$ | 4 | 16 | 1 | 8 | 8 | 32 |
| | cql weight | 0.001 | 0.5 | 0.001 | 1e-4 | 1 | 0.01 |
| | expectile $\tau$ | 0.7 | 0.99 | 0.7 | 0.7 | 0.7 | 0.7 |
| | discount $\gamma$ | 0.99 | 0.99 | 0.99 | 0.99 | 0.99 | 0.99 |
| PPO | model | gpt2-medium, gpt2-medium, gpt2-xl | gpt2-small | gpt2-small | gpt2-small | gpt2-small | gpt2-small |
| | lr | 1e-6 | 1e-6 | 5e-6 | 1e-5 | 1e-5 | 3e-5 |
| | rollouts | 2048 | 512 | 4000 | 1024 | 512 | 512 |
| | batch size | 128 | 128 | 128 | 128 | 128 | 32 |
| | GAE $\lambda$ | 0.95 | 0.95 | 0.95 | 0.95 | 0.95 | 0.95 |
| | discount $\gamma$ | 0.99 | 0.99 | 0.99 | 0.99 | 0.99 | 0.99 |
| | KL coef. | 0.01 | 0.1 | 0.01 | 0.01 | 0.01 | 0.001 |
| | clip range | 0.2 | 0.2 | 0.2 | 0.2 | 0.2 | 0.2 |
| | BC loss weight | 0 | 0 | 0 | 0 | 0 | 10 |

Table 3: Hyperparameters for baseline experiments.

| alg. | BC | % BC | MC Return | ILQL | Online PPO | Online % BC | GPT4 | Human |
|---|---|---|---|---|---|---|---|---|
| FO Maze | -72.1 | -56.4 | -48.1 | -6.97 | -37.7 | -71.7 | -39.7 | |
| PO Maze | -79.5 | -82.9 | -80.3 | -52.9 | -91.7 | -79.5 | -88.0 | |
| FO Text-Nav | 0.39 | 0.54 | 0.63 | 0.88 | 0.81 | 0.62 | 0.52 | |
| PO Text-Nav | 0.25 | 0.49 | 0.58 | 0.76 | 0.80 | 0.53 | 0.21 | |
| Wordle | -2.81 | -2.85 | -2.16 | -2.04 | -2.63 | -2.15 | -5.42 | - |
| Chess | -22.3 | -56.5 | -28.2 | -21.4 | -16.0 | -22.3 | -81.3 | |
| Endgames | 0.112 | -0.439 | 0.588 | 0.452 | 0.814 | 0.112 | -22.87 | |
| 20Qs | -16.0 | -14.6 | -13.9 | -14.2 | -14.9 | -16.8 | -13.0 | |
| Guess | -17.0 | -15.2 | -11.2 | -12.5 | -15.1 | -19.2 | -10.13 | |
| Car | 44.5 | 54.8 | 57.2 | 46.3 | 50.5 | | | |

Table 4: Raw statistics for all tasks. In the main paper, the statistics are normalized. Refer to Table 9

| | Reward Min Score | Dataset Average Score | Reward Max Score |
|---|---|---|---|
| FO Maze | -101 | -83 | -6.84 |
| PO Maze | -101 | -83 | -25.75 |
| F0 Text-Nav | 0 | 0.26 | 1 |
| PO Text-Nav | 0 | 0.26 | 1 |
| Wordle | -6 | -4.12 | -1.94 |
| Chess | -401 | 0.21 | 1 |
| Endgames | -1 | 0.586 | 1 |
| 20Qs | -20.0 | -17.3 | -12.6 |
| Guess | -20.0 | -18.8 | -8.56 |
| Car | 0 | | |

Table 5: In this table we report the minimum, dataset average, and maximum reward used to normalize the results in Table 4 tp get Table 9.

### E.3 CHESS ENDGAMES

To evaluate the chess agent in endgame positions, we select 645 positions not contained in the training dataset and which are not trivially solvable. By trivially solvable, we mean a position which could be solved by stockfish in one to four moves. In order to check this, we use Stockfish's evaluation tools to select positions which are a mate in 15 or greater. We then have the chess agent play one game from each position of these positions and keep these positions fixed for evaluation purposes. In this case we consider filtered BC to be training BC on all of the trajectories which ended in a victory.

As we can see in the table above, PPO Online significantly outperforms all of the other methods. To investigate whether PPO Online's performance is simply due to dataset collected, we fine-tune our BC agent on the PPO Online dataset. We do ablations where the data used for training is from the last 50, 25 and 10 rounds of data collection for the PPO policy. We choose to do this ablation because we expect the quality of the PPO policy performance increases in the later rounds of data collection.

### E.4 WORDLE

To evaluate Wordle, we rollout 4096 trajectories against the environment and report the average reward across all the rollouts.

## F BASELINE DETAILS

### F.1 MC DETAILS

The target for these heads is the discounted return-to-go:

$$R_t = \sum_{i=t}^{T-1} \gamma^{i-t} r_t \tag{2}$$

and we use MSE loss for the $Q$ head:

$$J(Q) = \mathbb{E}_{(s_t, a_t, r_{t:T-1}) \sim \mathcal{D}} \left[ (Q(s_t, a_t) - R_t)^2 \right] \tag{3}$$

$$\tag{4}$$

where $\mathcal{D}$ represents the dataset. In MC, $Q(s_t, a_t)$ represents how much more rewards the policy will get if it takes action $a_t$ at the state $s_t$ under some policy (in this case the policy that collected the dataset).

|  | BC | BC-Won | ILQL | MC Returns | PPO Offline | PPO Online |
|---|---|---|---|---|---|---|
| reward | -23.189 | -56.522 | -20.46 | -25.47 | -20.90 | -15.95 |
| percent illegal | 24.929% | 34.91% | 24.76 % | 25.64% | 23.05% | 21.96% |
| episode length | 51.01 | 92.02 | 47.96 | 53.44 | 48.69 | 44.19 |

Table 6: Results of chess agent in the full game positions against Stockfish Elo 1200.

|  | BC | % BC | MC | ILQL | PPO Offline | PPO Online |
|---|---|---|---|---|---|---|
| reward | 0.112 | -0.439 | 0.588 | 0.452 | -0.019 | **0.814** |
| percent victories | 26.233 | 26.419 | 69.3 | 56.7 | 28.37 | **88.4** |
| percent illegal | 0.967 | 2.717 | 0.692 | **0.66** | 0.925 | 0.722 |
| episode length | 12.923 | 23.477 | 11.92 | 14.6 | 25.24 | **8.38** |

Table 7: Comparison between the different baseline methods. The best performance is achieved by PPO Online with a 0.13 gap in performance between PPO Online and the next best-performing method of MC Returns. PPO Online attains overall the highest reward, but BC-Engine wins more frequently and MC Returns and ILQL make fewer illegal moves.

During rollout, when sampling, we perturb the base BC policy with the learned value-functions (Snell et al., 2022a). Let $\pi_\beta$ represent the policy trained with BC, and $\alpha$ represent a scalar multiplier, then:

$$\pi_{\mathrm{MC}}(a_t|s_t) \propto \pi_\beta(a_t|s_t)^{\alpha Q(s_t,a_t)} \tag{5}$$

### F.2 PPO DETAILS

**PPO Implementation Details**  Our PPO implementation uses a learned value function to estimate an advantage baseline. Our value function is fit using GAE (Schulman et al., 2018) value estimates and is implemented as a linear head on top of the same transformer as the policy. We apply clipping to both the importance weights and the value function, as is done in (Castricato et al., 2023). We also apply a KL penalty to the reward, as is standard for RLHF (Stiennon et al., 2022). On some of our tasks, we add a supervised learning, BC, loss term to the standard PPO loss to prevent the model in distribution; this is similar to the approach taken in Ouyang et al. (2022), which adds a pretraining loss term to the PPO loss to mitigate performance regressions on some benchmarks during RLHF.

**PPO Instabilities**  In some cases we observed training instabilities with PPO in which the policy's performance would increase for a little bit and then collapse (see Figure 5). We primarily observed this on our maze environment. We are uncertain what the cause of this dramatic instability is, and leave further investigation of this phenomenon to future work.

## G  SYMBOLIC MAZE ENVIRONMENT

Code to run experiments for the symbolic maze task can be found here: REDACTED. We ran online and offline Q-learning on this environment. The online Q-learning method is DQN and the offline

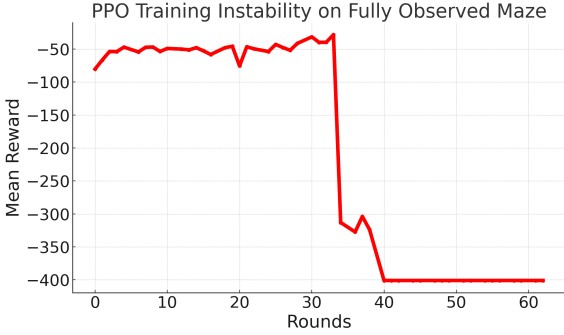

Figure 5: An example of an observed PPO training instability on the fully observed maze task.

|  | BC | Complete | Last 50 | Last 25 | Last 10 | PPO Online |
|---|---|---|---|---|---|---|
| reward | 0.112 | 0.201 | 0.17 | 0.189 | 0.235 | 0.814 |
| percent victories | 26.233 | 38.636 | 37.023 | 40.558 | 41.271 | 88.4 |
| percent illegal | 0.967 | 1.165 | 1.159 | 1.213 | 1.175 | 0.722 |
| episode length | 12.923 | 13.21 | 14.22 | 14.647 | 13.338 | 8.38 |

Table 8: Comparison between PPO Online and BC agents fine-tuned on the dataset collected by PPO during training. We chose to train on the complete PPO dataset, the last 50 rounds, last 25 rounds, and last 10 rounds of data collected. PPO Online performance still far surpassed performance of the BC agents trained on the PPO policy dataset. Furthermore, there is no substantive difference between training on the complete PPO dataset and the PPO dataset collected in the last 10 rounds.

Q-learning method is CQL. We found that Q-learning was able to perform optimally on the Maze task in both the online and offline setting. The results are as follows:

| alg. | DQN | CQL |
|---|---|---|
| Symbolic Maze | 100.0 | 100.0 |

Table 9: Results for CQL and DQN on the symbolic maze environment. These are normalized in the same way as results on LM RL tasks.

## H  GENERATED EXAMPLES FROM TRAINED MODELS

We show an example of a conversation between BC model (seller) and the oracle model (buyer):

```
===========
Preferred brands:  Lexus, 46 features, backup camera, blind
spot monitoring, sunroof.  Budget:  $90000.
===========
Seller:  Good afternoon!  How may I assist you today?
Buyer:  Hi there!  I'm looking for a Lexus SUV with a backup
camera, blind spot monitoring, and a sunroof.  Do you have
anything like that?
Seller:  Absolutely!  We have a few models that fit that
description.  Do you have a budget in mind?
Buyer:  Well, I'd like to stay around $90,000, but I'm
willing to go over that if I find the perfect car.
Seller:  Great!  Let me show you our selection.  This Lexus
SUV here has all the features you're looking for.  Have a
seat inside and take a look.  Buyer:  Wow, this is really
nice!  How much does it cost?  Seller:  The MSRP for this
model is $87,000, but we can work with you on the price.
Buyer:  Hmm, that's a bit high for me.  Do you have any
other options?
Seller:  Of course!  We have this similar model here for
$80,000.  It's a great car and has all the features you're
looking for.
Buyer:  That's more in my budget.  Let me take a look
inside.
Seller:  Absolutely!  And if you're interested, we could
also discuss financing options.
Buyer:  I'm definitely interested in financing.  How does
that work here?
Seller:  Well, we have a few options available.  We can work
with your bank or credit union, or we can discuss financing
through our dealership.  We also have some special offers
and promotions available that may be of interest to you.
Buyer:  That sounds good.  I definitely want to make sure
I'm getting the best deal possible.
Seller:  Of course!  We want to make sure you're completely
```

```
satisfied with your purchase.  And this car here is a great
value for the features you're getting.  Buyer:  You know
what, I think I'm sold!  Let's do this.
Seller:  Wonderful!  I just need to get some paperwork
started and we can get the financing process going.
Congratulations on your new car!
```

We show a conversation between the MC model (seller) and the oracle (buyer):

```
===========
Preferred brands:  Audi, Sunroof/moonroof, Leather seats,
Heated seats.  Budget:  $70000
===========
Seller:  Good morning, how can I assist you today?
Buyer:  I'm looking to buy an Audi Sedan.
Seller:  Certainly, we have a range of Audi Sedans
available.  What features are you interested in?
Buyer:  I want a car with a sunroof/moonroof, leather seats,
and heated seats.
Seller:  I understand.  Let me check our inventory for Audi
Sedans with those features.  (Pause)
Seller:  We have a few options that meet your requirements.
The prices range from $75,000 to $85,000.
Buyer:  My budget is $70,000.  Can you offer a lower price?
Seller:  I'm sorry, but we don't offer any discounts.  The
prices are as listed.
Buyer:  I really can't go over my budget.  Is there anything
you can do?  Seller:  I apologize for the inconvenience,
but the prices are fixed.  We have quality Audi Sedans that
meet your preferences.  Buyer:  (Sighs) Okay, let's say I
consider going slightly over my budget.  How much would it
be?  Seller:  The closest option to your budget is priced
at $75,000.  Buyer:  Hmm, that's not bad.  Can you go any
lower?
Seller:  I'm sorry, but I can't lower the price.
Buyer:  Alright, I'll take it.  What's the final price?
Seller:  The final price is $75,000.
Buyer:  Okay, let's proceed with the purchase then.
```

# I  HUMAN EVALUATION EXAMPLES

```
You are playing a game of twenty questions.  You can ask
20 yes-no questions to determine the identity of an object
chosen by an oracle.  Each turn, you can ask a question and
receives a "Yes" or "No" as the answer.  You are smart, so
you will ask the question that will narrow down the possible
objects as much as possible.  Don't get stuck on one idea
and try to branch out if you get stuck.

Human:  Does the item belong indoors?
Oracle:  No.
Is the item used for playing sports?
Oracle:  Yes.
Human:  Is the item a ball?
Oracle:  Yes.
Human:  Do you throw the item?
Oracle:  Yes.
Human:  Do you bounce the item?
Oracle:  No.
Human:  Is it a baseball?
Oracle:  No.
Human:  Is it a football?
Oracle:  Yes.
```

## J    RATING EXAMPLES FOR NATURALNESS

We conducted a user study where we provided participants with 18 examples to annotate on a scale from 1 to 5, where 1 represents the least natural and 5 represents the most natural conversation. There were 9 examples that were from GPT, and 9 examples that were generated from one of our models (MC Returns). We showed them examples from 20 Questions, Guess My City, and Car Dealer tasks. We found the following ratings below. Note that each element shows the percentage for the particular label. We found that participants felt conversations from GPT and the MC model where equally natural of 55.56% and 58.53% respectively.

| Label | Percentages for GPT | | | Percentages for MC Model | | |
|---|---|---|---|---|---|---|
| | Car Dealer | Guess City | 20 Questions | Car Dealer | Guess City | 20 Questions |
| 1 - Not Natural | 4.44% | 13.33% | 24.44% | 8.89% | 11.11% | 22.22% |
| 2 - Slightly Unnatural | 24.44% | 35.56% | 31.11% | 17.78% | 40.00% | 24.44% |
| 3 - Neutral/Natural | 20.00% | 24.44% | 15.56% | 17.78% | 20.00% | 8.89% |
| 4 - Quite Natural | 24.44% | 22.22% | 13.33% | 31.11% | 20.00% | 35.56% |
| 5 - Very Natural | 26.67% | 4.44% | 15.56% | 24.44% | 8.89% | 8.89% |

Table 10: User study of humans rating conversations from GPT and from our MC model for three tasks: Car Dealer, Guess City, and 20 Questions.

| Label | GPT | MC Model |
|---|---|---|
| 1 | 14.07% | 14.07% |
| 2 | 30.37% | 27.41% |
| 3 | 20.00% | 15.56% |
| 4 | 20.00% | 28.89% |
| 5 | 15.56% | 14.07% |
| Sum ($\geq 3$) | 55.56% | 58.52% |

Table 11: Average of percentage ratings for three tasks for GPT and for our MC Model

