# OpenReview forum: "LMRL Gym: Benchmarks for Multi-Turn Reinforcement Learning with Language Models"
_ICLR.cc/2024/Conference — Submitted to ICLR 2024_

### Official Review · Reviewer_fFLM · 2023-10-27

**Soundness:** 3 good
**Presentation:** 3 good
**Contribution:** 2 fair
**Rating:** 6
**Confidence:** 4

**Summary:**

This paper introduces **LMRL-Gym**, consisting of **8 tasks** ranging from simple navigation (Maze) to strategy games (Chess) to negotiation (Car Dealer).

They also provide a **research toolkit** for practitioners to get started with multi-turn RL for LLMs.

This benchmark is created for evaluating RL algorithms in multi-turn language-based interaction tasks using LLMs as agents and simulators.

It provides a research framework and discusses the key challenges and capabilities involved in training RL algorithms for LLMs.

The goal is to advance the development of more effective RL methods for language-based interactions, including complex decision-making scenarios and conversational interactions, with a focus on accessibility for researchers with varying computational resources.

**Strengths:**

1. Present a useful **dataset**: It proposed a novel and significant challenge in the field of reinforcement learning and natural language processing, focusing on the application of RL algorithms to Large Language Models (LLMs) for multi-turn language-based interactions.

2. The **Data Generation Approach** can be used for collecting more data.

3. They designed **8 tasks** in the LMRL-Gym benchmark to **evaluate** the core capabilities that RL can enable in large language models. Their evaluation shows the promise of RL in several tasks, with further room for improvement with a push for better methods.

4. It provides a **clear explanation** of the motivation behind benchmarking RL algorithms for LLMs and the need for such an evaluation framework.

**Weaknesses:**

1. LLM with RL/PPO in this complicated task is very sensitive to the hyper-parameters. I did not find the analysis or report on these.

2. The Data Generation Approach is a little disappointing. I did not find the details that can replicate this process or any validation of the data. How can I make sure these data can be used to evaluate RL?

3. The baseline results are not convincing enough. PPO Instabilities in GPT-2-small are very obvious. I do not think the results in the benchmark should only use GPT-2-small. Any technique you use here to overcome it?

**Questions:**

1. Any details about the hyperparameters for baselines? Thanks.

2. Any validation of the data generation process? How to make sure the game/task you generate is POMDP?

3. How to replicate the data generation process?

4. How did you evaluate the quality of your data?

5. I think I might miss it. You use the data from the GPT-3.5 and create several games for the 8 tasks?

---

> ### Author Response · Authors · 2023-11-19
>
> We thank the reviewer for their feedback. We would like to address your questions and concerns by 1) further explaining our data generation process 2) explaining how to replicate the data generation process, and 3) directing your attention to Appendix D on hyperparameters. We address your comments point by point below:
>
> > “You use the data from the GPT-3.5 and create several games for the 8 tasks?
>
> Yes, for our three dialogue tasks, we do the following: 1) Generate data with GPT-3.5 based on specific prompts (for an example for Twenty Questions, see Appendix, Section C.6) 2) Distill this to two GPT2 models one for the agent, one for the environment 3) Use these distilled models to generate a new dataset that we will use for training our algorithms. See Section 4.4 of our paper for further information on data generation and reference Figure 2.
>
> > “How to replicate the data generation process?”
>
> The data generation process can be replicated by using the GPT2 checkpoints provided to regenerate the data. We also provide our prompts and data generation scripts in our code base, for which we have provided an updated version.
>
> > “How did you evaluate the quality of your data?”
>
> We evaluate the quality of our data by 1) inspecting the generations for signs of problems 2) performing a human evaluation where humans interact with the simulator model that generated the data. If humans are able to successfully interact with the models, it is a clear signal that our data is also natural and contains the desired properties. Additionally, we have found better performance from our algorithms compared with the BC models, signaling that the data the simulator was trained on is providing a useful signal for the improvement of RL algorithms.
>
> > “Any validation of the data generation process? How to make sure the game/task you generate is POMDP?”
>
> To validate the data generation process, we report statistics as can be seen in Table 1. We also provide further details on the dataset generated in Appendix C. For the text-games we ensure it is a POMDP, by creating environments such that the next state is only dependent on the previous state and action. For dialogue tasks, the state can be thought of as a concatenation of all previous utterances. With this definition, this situation will also be a POMDP.
>
> > “LLM with RL/PPO in this complicated task is very sensitive to the hyper-parameters. I did not find the analysis or report on these….Any details about the hyperparameters for baselines?”
>
> We have reported our hyperparameters for training the baselines in Appendix D.
>
> > “The baseline results are not convincing enough. PPO Instabilities in GPT-2-small are very obvious. I do not think the results in the benchmark should only use GPT-2-small. Any technique you use here to overcome it?”
>
> We overcome instabilities in PPO by 1) increasing the number of rollouts 2) tuning the KL coefficient and 3). Regarding using GPT2-small, we were limited by compute and simply cannot train 7B+ models on all these settings. However, for the models we use as the environment, we have used GPT2-XL for the Car Dealer task.
>
> We thank the reviewer for their helpful feedback in improving the paper, and hope that you find these reasons convincing in raising your score!

---

> > ### Author Response · Authors · 2023-11-20
> >
> > Let us know if you have any further questions that we may clarify!

---

> > > ### Author Response · Authors · 2023-11-21
> > >
> > > Let us know if there is anything else we can elaborate on for the last day of discussion period :)

---

> > > > ### Comment · Reviewer_fFLM · 2023-12-01
> > > > **Thanks for your efforts. I have raised the rating to 6.**
> > > >
> > > > Thanks. I really appreciate your efforts. I have raised my rating.

---

### Official Review · Reviewer_6Fq6 · 2023-10-30

**Soundness:** 2 fair
**Presentation:** 3 good
**Contribution:** 3 good
**Rating:** 5
**Confidence:** 4

**Summary:**

The authors propose a set of environments that can be used to assess the ability of RL algorithms to fine tune and/or train LLMs. The creation of the benchmark is intended to enable evaluating different RL algorithms on multi-turn, language-based tasks. All environment state and action spaces are expressed solely in language. The set of tasks include a maze, a house-based maze, World, chess, chess endgames, twenty questions, guess my city, and car dealer. The environments are selected/designed to evaluate different capabilities that are expected from a LLM, such as common sense reasoning, credit assignment, reasoning under uncertainty, information seeking behaviors, and trajectory stitching. Where an environment requires dialogue-like interactions (i.e. 20 questions, guess my city, and car dealer), an GPT2 train on environment-specific data is used to provide the environment responses. The environments are designed not to assess the ability of LLM to communicate with humans, but to assess how well they can solve different reasoning tasks.

**Strengths:**

- The paper is well written and easy to follow
- The benchmark provides a way to quantitatively measure different LLM reasoning abilities, specifically for several environments, open vocabulary tasks are assessed.

**Weaknesses:**

- The authors motivate that a large part of the goal is to assess how well RL algorithms perform on language-based tasks, but they do not provide evidence that RL algorithm performance differences on non-language tasks does not correlate with algorithm performance on language-based tasks. As the environments are on the contrived side, it is not clear they address the challenges associated with training LLMs.
- While the state and action spaces are language-based, it is not clear the extent to which some of the environments assess performance on language-based tasks, such as the maze, chess, and chess endgames. While the environments assess reasoning abilities, they do not assess the full complexity of reasoning abilities a LLM needs nor abilities that are key to LLM success. The reasoning abilities are more general abilities we would want from a RL-train agent. For example, it is important for LLMs to be factual and harmless, which means correcting behavior learned during pretraining.
- It is not clear what safeguards are in place to prevent hacking of the LLM that is part of the environment.

**Questions:**

- To what extent does the policy's actions on the environments like 20Qs, car dealer, and guess my city look like reasonable sentences?
- For some of the environments the specific skills addressed are called out. It would be great to highlight the assessed skills for each environment.
- Are there other LLM-specific policy learning algorithms that can be assessed?
- To what extent are the performance differences between RL algorithms proportional to their differences on non-language based tasks?

---

> ### Author Response · Authors · 2023-11-19
>
> We thank the reviewer for their feedback. We've addressed the main issues raised in your review by: (1) rewriting Section 4.1 of the paper to improve the definitions of the RL capabilities and how they apply to each task; (2) conducting a user study to test the naturalness of conversation from our simulators; (3) providing further examples of conversations from our trained models against the LLM simulator. We address your comments point by point below:
>
> “While the state and action spaces are language-based, it is not clear the extent to which some of the environments assess performance on language-based tasks, such as the maze, chess, and chess endgames. While the environments assess reasoning abilities, they do not assess the full complexity of reasoning abilities a LLM needs nor abilities that are key to LLM success. The reasoning abilities are more general abilities we would want from a RL-train agent. For example, it is important for LLMs to be factual and harmless, which means correcting behavior learned during pretraining.”
>
> It's definitely a valid point that not all the tasks test performance on realistic language-based tasks, though some do. However, each of the tasks in the benchmarks serves a different purpose. Some tasks (Guess My City, Car Dealer) aim to evaluate tasks with realistic natural language. Some tasks aim to test specific RL properties without the complexities of realistic language, while others focus on complex language. Algorithms developers would be expected to evaluate their methods on the totality of all the tasks. We've expanded Section 4.3 to clarify this. We do not believe that this is a weakness, as our benchmark does include a number of tasks with realistic language, and it seems valuable to test a variety of RL algorithm capabilities in isolation (and comparatively less valuable to have redundant tasks that all test similar things). However, we hope that the revised discussion in Section 4.3 more clearly explains this rationale.
>
> > “To what extent does the policy's actions on the environments like 20Qs, car dealer, and guess my city look like reasonable sentences?”
>
> We find that all of the policy’s actions on the environment appear to be reasonable sentences. We have verified this by looking at the generated datasets and interacting with the environments ourselves, as well as performing a user study of having humans evaluate if the text used to train the simulator from GPT and from one of our models (MC returns) was natural on a scale of 1-5 (where 5 is most natural). On average, we found humans to say the text from GPT-4 was 55.56% natural, and from our model was 58.51% natural. We provide detailed statistics and explanations of the user study evaluating naturalistic language in our Appendix, Section J. We are also currently conducting a larger human evaluation study of humans interacting with our models as a baseline, and provide partial results in to Table 2 in our paper with some discussion. We also show sample interactions of our BC and MC policy model with the oracle in Appendix, Section H for the Car Dealer Task.
>
> > “Are there other LLM-specific policy learning algorithms that can be assessed?”
>
> Yes, since our submission, we have performed a new evaluation experiment by few-shot prompting GPT-4. This is a LLM-specific policy learning algorithm because only LLMs can be few-shot prompted in this way. In addition, we have run an ablation with Online Filtered BC, which is an algorithm that collects data using the current policy and selects the most successful trajectories for fine-tuning. This is an algorithm that is often used in RLHF pipelines. In general, we have found that these evaluations did not outperform our previous baselines. We include these results in our updated results section.
>
> > “For some of the environments the specific skills addressed are called out. It would be great to highlight the assessed skills for each environment.”
>
> We agree that it is important to highlight the capabilities enabled by every task. As per your suggestion, we have updated our draft with Section 4.2.1 and an additional Figure 3 to more clearly highlight which tasks enable which RL capability and why.  For example, the Maze and Text-Nav tasks contain both partially observed and fully observed versions to highlight the impact of partial observability. In addition, the Text-Nav task is very similar to the Maze task, but has the added complexity of learning to generate realistic text.>

---

> > ### Author Response · Authors · 2023-11-19
> >
> > > “To what extent are the performance differences between RL algorithms proportional to their differences on non-language based tasks?”
> >
> > This is an excellent question and one that we investigated by creating a non-text based version of the maze task. We found that simple online and offline Q-learning was able to get an optimal score on the maze. Therefore, performance symbolic maze is comparable to the fully observed Maze task. However, on the partially observed Maze task, the language based methods perform significantly worse. This highlights room for improvement on dealing with partial observability in RL with language. We have added these details in our updated revision in the Appendix, Section G.
> >
> > > "It is not clear what safeguards are in place to prevent hacking of the LLM that is part of the environment.”
> >
> > We thank the reviewer for this question. We clarify the methodology with which we generated our datasets to train our simulators. As shown in Figure 2, we train simulators that serve as an “oracle” for the task, which defines the ground truth environment. In most cases, our simulator is performing an easier task than the agent, which does not require strategic reasoning. Additionally, our tasks are designed such that any noise in our environment will not necessarily break the task, but merely add an additional inference challenge for the agent. For example, the role of the oracle in the Twenty Questions task is to provide objective yes/no answers to questions about the object, and in Guess My City, to provide more open-ended information about a query on the city. OpenAI’s GPT-3.5 has been shown to be able to generate reasonable questions and answers when used out of the box, which is why we leveraged it to collect our initial dataset. We have provided prompts that we use to generate the data to train our oracle models in our Appendix, Section C.6, and snippets below to show our thought process to maintain high accuracy.
> >
> > With respect to the Car Dealer task, we spent a considerable effort to ensure diversity in the responses of sellers, by providing different desired brands, features, classifications (i.e. car or truck), and budgets in our prompting to generate the datasets. We provide a sample of conversation between the oracle model and MC returns vs. oracle and the BC model to illustrate in our Appendix, Section H.
> >
> > > “The authors motivate that a large part of the goal is to assess how well RL algorithms perform on language-based tasks, but they do not provide evidence that RL algorithm performance differences on non-language tasks do not correlate with algorithm performance on language-based tasks. As the environments are on the contrived side, it is not clear they address the challenges associated with training LLMs.”
> >
> > Refer to our response regarding comparison of performance of the non-text based version of the maze task with the text-based version, details of which we have included in Appendix, Section G. Additionally, we have expanded Section 4.3 to explain the evaluation capabilities of each of our tasks.
> >
> > We thank the reviewer for their helpful feedback in improving the paper, and hope that you find these reasons convincing in raising your score!

---

> > > ### Author Response · Authors · 2023-11-20
> > >
> > > Let us know if you have any further questions that we may clarify!

---

> > > > ### Comment · Reviewer_6Fq6 · 2023-11-21
> > > > **Response to Authors**
> > > >
> > > > Thank you for your response.
> > > >
> > > > You call out that there is space for improvement on the partially observable MazeGame for the LLM. It is not clear to me, did you evaluate a partially observable non-language-based condition?

---

> > > > > ### Author Response · Authors · 2023-11-22
> > > > >
> > > > > Thank you for your thoughtful response. We want to clarify our initial response and say that we did not evaluate a partially observed version for a non-language-based MazeGame.  Between the fully-observed and partially-observed variations of the language-based version, we observe a significant performance gap.
> > > > >
> > > > > We set up a non-language-based partially observed policy that outperforms the learned policy significantly. We believe that this signals room for improvement. This policy works by computing a probability distribution over possible locations in the maze and maximizes information gain when selecting an action.
> > > > >
> > > > > The purpose of the fully-observed non-language-based MazeGame was to set an upper bound on both sample efficiency and performance for the language-based versions. We find it interesting that although ILQL matches the performance of CQL on the fully observed non-language task, MC Returns, PPO, and BC do not. On the other hand, MC Returns performs better than ILQL on other, more complicated tasks. This discrepancy highlights room for improvement for the RL baselines.

---

### Official Review · Reviewer_BtsG · 2023-11-01

**Soundness:** 2 fair
**Presentation:** 2 fair
**Contribution:** 2 fair
**Rating:** 6
**Confidence:** 4

**Summary:**

This paper proposes a new benchmark of reinforcement learning on language models in multi-turn scenarios. The proposed benchmark called LMRL-Gym includes 8 different tasks, covering open-ended dialogue and text games. These tasks require the five capabilities of LM that can be enabled by RL: complex decision making, complex language, credit assignment, partial observability, and trajectory stitching. The authors applied supervised fine-tuning (defined as behavior cloning (BC) in this paper), value-based offline-RL, and online PPO to form the baselines for the proposed benchmark. The results show that RL can improve BC and still leave room for improvement.

**Strengths:**

* This benchmark can be useful for comparison of RL algorithms on multi-turn text generation. This area is less explored compared with single-turn.
* The proposed benchmark considers diverse tasks from text game to open-ended conversation, and considers different dimensions of LM capabilities RL can perhaps help.
* Most of the writing is clear.

**Weaknesses:**

* The benchmark is artificial and quite less natural. Only two of the eight tasks (Car Dealer and Guess My City) is multi-turn conversation or conversational QA with more unbounded state and action space. All other tasks have obvious restrictions, especially, the Chess and Endgames tasks are not natural language generation to me.
* The experiments section can be improved by providing more analysis. For example, the authors can analyze which capability in Section 4.1 is easier or less likely to be improved. Also, the authors can consider how to disentangle the evaluation of capabilities in each task.
* Generated examples by the trained agents can be added. Although the authors have put emphasis on “our goal is not to utilize this approach to benchmark whether LLMs are good at talking to humans, but rather as a way to test RL algorithms with datasets that are sufficiently difficult and complex so as to gauge how effective they might be if they were then trained on data from real humans.“ in Introduction, it is good to see how relevant the obtained rewards and the conversation's naturalness are.
* While the paper claims to focus on LLM, the experiments are conducted on GPT2-small and medium (according to Table3), which may be controversial. I’m also confused by the contradiction in Section 7 and Table3. In Section 7, the authors say they “primarily trained and evaluated models with a maximum 1.5B parameters”, but Table3 shows the experiments are conducted on less than 355M parameters. Therefore, I’m wondering how large is the trained agent in Table2 actually?
* The writing of section 4.1 can be further improved. For now, I can guess the meaning from the terminologies themselves, but I will get confused from the content in each paragraph trying to explain the corresponding terminology.

**Questions:**

* Does the timestep in Table 1 mean for one response instead of for one token, and the length mean the number of turns in a conversation?  If so, the wording of timestep and length may be a bit confusing with the terminologies in single-turn text generation, where the timestep often refers to the step of generating a token and the length often refers to how many tokens per response.
* Why the avg length for 20Qs and Guess in Table 1 are negative?
* How many different state-action pairs exist in the dataset for offline training?
* Some details can be moved from Appendix B to section 4.3. For example, the desired capability for every task. For now some are left in the appendix.
* Typos:
  * Is the Twenty Questions task example in Figure 2 needed to exchange the labels of environment and agent? I guess it should be the environment to say yes or no, but not the agent.
  * There is no reference to table in Appendix C (which is mentioned in section 5.1). I guess Appendix C should mention Table 3 in it.
  * Do the authors actually mean “do NOT consider” in Section 2 - 2nd paragraph? The original sentence: “However, many of these works operate in the single-step bandit setting, and do consider multi-turn goal-directed tasks.”
  * Typo: Section 3.1 POMPDP -> POMDP

---

> ### Author Response · Authors · 2023-11-19
>
> We thank the reviewer for their feedback. We've addressed the main issues raised in your review by: (1) rewriting Section 4.1 of the paper to improve the definitions of the RL capabilities and how they apply to each task; (2) conducting a human evaluation study to test the naturalness of conversation from our simulators; (3) providing further examples of conversations from our trained models against the LLM simulator. We address your comments point by point below:
>
> > “The benchmark is artificial and quite less natural. Only two of the eight tasks (Car Dealer and Guess My City) is multi-turn conversation or conversational QA with more unbounded state and action space. All other tasks have obvious restrictions, especially, the Chess and Endgames tasks are not natural language generation to me.”
>
> It's definitely a valid point that not all of our tasks have realistic natural language, though some do. However, each of the tasks in the benchmarks serves a different purpose. Some tasks (Guess My City, Car Dealer) aim to evaluate tasks with realistic natural language. Some tasks aim to test specific RL properties without the complexities of realistic language, while others focus on complex language. Algorithm developers would be expected to evaluate their methods on the totality of all the tasks. We've expanded Section 4.3 to clarify this. We do not believe that this is a weakness, as our benchmark does include a number of tasks with realistic language, and it seems valuable to test a variety of RL algorithm capabilities in isolation (and comparatively less valuable to have redundant tasks that all test similar things). However, we hope that the revised discussion in Section 4.2.1 more clearly explains this rationale.
>
> > “Generated examples by the trained agents can be added. Although the authors have put emphasis on “our goal is not to utilize this approach to benchmark whether LLMs are good at talking to humans, but rather as a way to test RL algorithms with datasets that are sufficiently difficult and complex so as to gauge how effective they might be if they were then trained on data from real humans.“ in Introduction, it is good to see how relevant the obtained rewards and the conversation's naturalness are.”
>
> Thank you for this question. Rewards for the tasks are simple and intuitive, because they are based on success of selling the car or guessing the word. To address your concern regarding the naturalness of the conversation, we have provided a few examples in the Appendix, Section H. In addition, we have evaluated all of our tasks with human evaluators playing the games to assess how natural they are, and show an example in Appendix, Section I.
>
> > “While the paper claims to focus on LLM, the experiments are conducted on GPT2-small and medium (according to Table3), which may be controversial. I’m also confused by the contradiction in Section 7 and Table3. In Section 7, the authors say they “primarily trained and evaluated models with a maximum 1.5B parameters”, but Table3 shows the experiments are conducted on less than 355M parameters. Therefore, I’m wondering how large is the trained agent in Table2 actually?”
>
> Thank you for your question. This is a valid concern. We trained one model (Car Dealer) with GPT2-XL, and missed putting this into Table 2, which is now updated. We were limited by compute and simply cannot train 7B+ models on all these settings. However, for the models we use as the environment, we use GPT2-XL.
>
> > “The writing of section 4.1 can be further improved. For now, I can guess the meaning from the terminologies themselves, but I will get confused from the content in each paragraph trying to explain the corresponding terminology. … Some details can be moved from Appendix B to section 4.3. For example, the desired capability for every task. For now some are left in the appendix.”
>
> In order to make Section 4.1 clearer, we have improved the definitions of the RL capabilities. We have added Section 4.2.1 and an additional Figure 3 to more clearly highlight which tasks enable which RL capability and why.

---

> > ### Author Response · Authors · 2023-11-19
> >
> > > “Does the timestep in Table 1 mean for one response instead of for one token, and the length mean the number of turns in a conversation? If so, the wording of timestep and length may be a bit confusing with the terminologies in single-turn text generation, where the timestep often refers to the step of generating a token and the length often refers to how many tokens per response.”
> >
> > Yes, we acknowledge that this is confusing. By “timestep” we mean “one response by the agent” and length is “number of responses by the agent”. We have updated the caption in Table 1 and definitions in Section 3 to clarify this. As we are attempting to do something that has not been explored much before, working out the proper terminology is tricky. Thank you for your feedback!
> >
> > > “Why the avg length for 20Qs and Guess in Table 1 are negative?”
> >
> > The negative is indicative of the reward and was a typo for the average length, which has been fixed in the table.
> >
> > We have also corrected the typos the reviewer has found and updated our paper. We thank the reviewer for their helpful feedback in improving the paper, and hope that you find these reasons convincing in raising your score!

---

> > > ### Author Response · Authors · 2023-11-20
> > >
> > > Let us know if you have any further questions that we may clarify!

---

> > > > ### Author Response · Authors · 2023-11-21
> > > >
> > > > Let us know if there is anything else we can elaborate on for the last day of discussion period :)

---

> > > > > ### Comment · Reviewer_BtsG · 2023-11-23
> > > > > **Thank you for your responses!**
> > > > >
> > > > > I’m grateful to the authors for their response and paper revision. The authors have addressed most of my concerns, so I would raise my score from 5 to 6. I’m not giving a higher score since my major concern still exists -- the naturalness. The proposed LMRL-Gym is artificial compared to some existing LM tasks people are already working on using RL. Even the more natural tasks in it are also more limited than open-ended conversation. I understand that this can be a tradeoff for investigating diverse, desired capabilities of tasks. However, the naturalness can be a big difference between LM and other RL tasks.

---

### Official Review · Reviewer_2ASy · 2023-11-01

**Soundness:** 2 fair
**Presentation:** 2 fair
**Contribution:** 3 good
**Rating:** 5
**Confidence:** 4

**Summary:**

This paper introduces LMRL-Gym, a benchmark for evaluating reinforcement learning (RL) algorithms for training large language models (LLMs) on multi-turn tasks. The benchmark consists of 8 tasks spanning open-ended dialogue, strategic games, and tool use. The tasks are designed to test core capabilities like complex decision making, handling complex language, credit assignment, and partial observability. The authors benchmark training methods like offline RL algorithms (ILQL), online RL algorithms like PPO and supervised fine-tuning (BC) on their tasks.

**Strengths:**

- Establishes an accessible benchmark to drive progress on an important open problem - multi-turn RL for language models. This direction for development is timely and much needed. There are no systematic benchmarks available.
- Well motivated, it is easy to follow and clear in writing.
- Tasks are designed in a modular fashion, starting from simple unit tests and building up complexity.
This enables systematic testing of capabilities.
- Provides datasets, evaluation protocols, baseline implementations - lowers barriers to entry for future research.
- The authors make a clear distinction of the purpose of their paper: To test RL methods on sufficiently complex tasks with LLMs
- The authors have a good breadth of methods that they test with the
- Broad set of baseline algorithms are evaluated.

**Weaknesses:**

- The reasoning and generalization required on each task needs to be formalized more clearly. What inferences must the model make? How different are test vs training distributions? This formalization will further help understand how the complexity of tasks varies across different tasks.
- An alternative to improve interaction and performance at tasks is to use prompting and factoring of LLM calls. I found a discussion on prompting vs RL training as alternatives to each other missing.
- For complex tests,  20 questions, guess the city, car dealership where the model relies on GPT-3.5 for the initial test and GPT-2 for the env, there need to be evaluations of the text produced by the models.
    - The dependent measure would be non-sensical if the fixed model is not ‘rational’ at interacting with the model being tested.
    - GPT-3.5-turbo could be inconsistent. And the finetuned GPT-2 model could be too!
    - The measure cant be trusted if the environment itself is inconsistent.
    - This further leads to the question if the environment is easily prone to hacking! There should at least be qualitative trajectories that show the measure and the environment are reasonable. A trained model could figure out a weakness of the eval gpt-2 model to gain a higher reward.
- In wordle, i’d love to see a discussion of how the authors deal with the tokenization. Since the models are bad at tasks where individual characters need to be tokenized.
- Why did the authors not choose to start with human data in domains where that was available? Eg: Craigslist dataset, Deal or No Deal
- In the car dealership case App B8, a negotiation setting, the values of the interacting agents are much more interesting than their personalities. I would have loved to see a negotiation setting where the values of the buyers are varied and not just 3 discrete buckets that buyers and sellers are put into.
- There are no error bars for the RL methods. The authors should report least report 5 runs.
- The social/ethical implications section is incomplete: Dual use concerns around persuasion, manipulation, and addictive engagement should be discussed given interactive RL.

**Questions:**

Questions and suggestions gave been included in the weaknesses above.

---

> ### Author Response · Authors · 2023-11-19
>
> We thank the reviewer for their feedback. We've addressed the main issues raised in your review by: (1) including a comparison between few-shot prompting GPT-4 and our RL baselines; (2) rewriting our paper to more clearly articulate the capabilities being tested; (3) providing evidence in the form of a user study that our tasks reflect naturalistic language.  We address your comments point by point below:
>
> > “An alternative to improve interaction and performance at tasks is to use prompting and factoring of LLM calls. I found a discussion on prompting vs RL training as alternatives to each other missing.”
>
> As per your suggestion, we have included an additional experiment for few-shot prompted GPT-4 against all of the tasks. We have found that prompted GPT-4 was not able to outperform our baselines on the tasks. We have updated Section 6 of our paper with further details and discussion.
>
> > “The reasoning and generalization required on each task needs to be formalized more clearly. What inferences must the model make?”
>
> In Section 4, we delineate the capabilities enabled by RL, including strategic decision making, complex language, credit assignment, partial observability, and trajectory stitching. We have added Appendix A to formalize the reasoning required for each task. To address your question, we have added Section 4.2.1 and an additional Figure 3 to more clearly highlight which tasks enable which RL capability and why.
>
> > “For complex tests, 20 questions, guess the city, car dealership where the model relies on GPT-3.5 for the initial test and GPT-2 for the env, there need to be evaluations of the text produced by the models.”
>
> To address the concerns regarding the evaluation of text produced by the models, we have performed an additional evaluation of having humans evaluate if the text from both GPT and from one of our models (MC returns) was natural on a scale of 1-5 (where 5 is most natural). On average, we found humans to say the text from GPT-4 was 55.56% natural, and from our model was 58.51% natural. We provide detailed statistics and explanations of the user study evaluating naturalistic language in our Appendix, Section J. We are also currently conducting a larger human evaluation study of humans interacting with our models as a baseline, and provide partial results in our updated paper.
>
> Additionally, we would like to clarify the methodology with which we generated our datasets to train our simulators. As shown in Figure 2, we train simulators that serve as an “oracle” for the task, which defines the ground truth environment. In most cases, our simulator is performing an easier task than the agent, which does not require strategic reasoning. Additionally, our tasks are designed such that any noise in our environment will not necessarily break the task, but merely add an additional inference challenge for the agent. For example, the role of the oracle in the Twenty Questions task is to provide objective yes/no answers to questions about the object, and in Guess My City, to provide more open ended information about a query on the city. OpenAI’s GPT-3.5 has been shown to be able to generate reasonable questions and answers when used out of the box, which is why we leveraged it to collect our initial dataset. We have provided prompts that we use to generate the data to train our oracle models in our Appendix, Section C.6, and snippets below to show our thought process to maintain high accuracy.
>
> With respect to the Car Dealer task, we spent a considerable effort to ensure diversity in the responses of sellers, by providing different desired brands, features, classifications (i.e. car or truck), and budgets in our prompting to generate the datasets. We provide a sample conversation between the oracle model and MC returns vs. oracle and the BC model to illustrate in our Appendix, Section H.
>
> > “Why did the authors not choose to start with human data in domains where that was available? Eg: Craigslist dataset, Deal or No Deal”
>
> Regarding why we did not choose to start with human data in domains that were available for the following reasons: 1) These datasets do not come with simulators that are faithful and produce data that is similar to the dataset 2) These datasets are not designed for RL and do not have clear and intuitive reward functions. Past works required various special choices to use them, for example, Verma et. al replaced the prices in the Craiglist dataset dialogue and used GPT to generate templates to substitute such prices, as they are not very realistic. Overall, we have found that these datasets do not encourage strong bargaining capabilities.
>
> Works Cited:
>
> [1] Verma, Siddharth, et al. "Chai: A chatbot ai for task-oriented dialogue with offline reinforcement learning." arXiv preprint arXiv:2204.08426 (2022).

---

> > ### Author Response · Authors · 2023-11-19
> >
> > > “The social/ethical implications section is incomplete: Dual use concerns around persuasion, manipulation, and addictive engagement should be discussed given interactive RL.”
> >
> > We thank the reviewer for raising this important concern and will add the following to this section:
> >
> > This work aims to develop a benchmark for the advancement of research in reinforcement learning and LLMs. In considering the social and ethical implications of interactive RL, we acknowledge and recognize the dual use implication of this research, particularly centered around developing LLM simulators that could perform persuasion, manipulation, and addictive engagement of users at a large scale. The optimization processes employed by such algorithms, which aim to maximize certain objectives, raise ethical considerations when the optimized outcomes may prioritize system goals over user safety and alignment with human values. We have designed our datasets and reward functions such that prioritize fairness and human-aligned outcomes. By incorporating these considerations when designing our framework, we aim to encourage the development of reinforcement learning models and LLMs that not only excel in performance but also adhere to ethical standards, mitigating the potential for nefarious persuasion or manipulation.
> >
> > > “How different are test vs training distributions?”
> >
> > To address your question regarding how different are the training and test distributions - the training and evaluation distributions are the same for all of our tasks, and depending on the task, we perform a different train/test splits (i.e. 90% training, 10% evaluation for Wordle task). We have provided additional examples in the Appendix, Section C.8 of our training and evaluation sets for the Car-Dealer task.
> >
> > > “In Wordle, I’d love to see a discussion of how the authors deal with the tokenization. Since the models are bad at tasks where individual characters need to be tokenized.”
> >
> > We have added additional details regarding experiments for tokenization in Wordle in our Appendix, Section C.3.
> >
> > > “There are no error bars for the RL methods. The authors should report at least report 5 runs.”
> >
> > We did not include training error bars for our methods due to compute constraints, as it may take up to 10 days to train these algorithms. However, we did do multiple runs for many tasks with different hyperparameters, which we show in the Appendix. Additionally, we do multiple rollouts for our evaluation, which is also displayed in the Appendix.
> >
> > We thank the reviewer for their helpful feedback in improving the paper. We hope that you find these reasons convincing and consider raising your score!

---

> > > ### Author Response · Authors · 2023-11-20
> > >
> > > Let us know if you have any further questions that we may clarify!

---

> > > > ### Author Response · Authors · 2023-11-21
> > > >
> > > > Let us know if there is anything else we can elaborate on for the last day of discussion period :)

---

> > > > > ### Comment · Reviewer_2ASy · 2023-11-23
> > > > > **Response to the authors**
> > > > >
> > > > > I thank the authors for their detailed response.
> > > > >
> > > > > A few things are essential which would help the paper be much stronger:
> > > > > - Larger human study
> > > > > - Several runs for RL algorithms
> > > > > - Proving that the environment is robust to simple reward hacking. Especially when there is a finetuned model that dictates the dependent measure.
> > > > > - Having more held-out splits for generalization
> > > > >
> > > > > I raise my score, but I think a second round of review will make the paper quite interesting and strong. I wish the authors the best, looking forward!

---

### Meta-Review · Area_Chair_moGQ · 2023-12-06

**Metareview:**

The paper introduces LMRL-Gym, a benchmark for evaluating reinforcement learning algorithms in training LLMs for multi-turn tasks. It encompasses eight diverse tasks, such as open-ended dialogue and strategic games, designed to assess desirable capabilities for RL algorithms, including strategic decision-making and credit assignment. The benchmark evaluates various training methods, including offline and online RL algorithms and supervised fine-tuning. As such, LMRL-Gym could significantly contribute to the community. However, after the reviewer-AC discussion and reviewing the authors' responses, several reviewers remain unconvinced due to the following limitations:

1. The experiments in the paper are relatively lightweight for a benchmarking paper. For instance, there are no error bars in the results, although error bars are crucial in reporting RL results. The authors should ideally report results from at least five runs. While the author response mentioned computational challenges in adding such experiments, this absence is a significant drawback for a paper about benchmarking. The reviewers also found the human study to be relatively small and suggested several improvements to the experimental section.
2. Reward Hacking: It is unclear what safeguards are in place to prevent hacking of the LLM that is part of the environment. Generally, there should be more attempts to demonstrate the environment's robustness against simple reward hacking, especially involving a fine-tuned model that dictates the dependent measure. Although the authors responded to one of the two reviewers who raised concerns about reward hacking, the reviewers remain unconvinced. The next version of the paper should directly address this issue.
3. LMRL-Gym’s focus on artificial tasks: More a restriction than a flaw, LMRL-Gym focuses significantly more on artificial tasks, like Mazes and Chess, than on natural ones, such as open-ended dialogue. Only two tasks (Car Dealer and Guess My City) involve natural multi-turn conversation or conversational QA with more unbounded state and action spaces, and they are domain-specific. This focus makes LMRL-Gym quite artificial compared to other LM-related tasks that the community is already working on using RL.

Considering these limitations and the continued skepticism of some reviewers, I recommend rejecting the paper. However, the reviewers and I generally agree that this work has great potential. If the authors incorporate the reviewers' suggestions, this could become a strong and interesting paper.

**Justification For Why Not Higher Score:**

Concerns about the experiments and reward hacking (weaknesses 1 and 2).

**Justification For Why Not Lower Score:**

N/A

---

### Decision · Program_Chairs · 2024-01-16

Reject